# YTHDC1 promotes postnatal brown adipose tissue development and thermogenesis by stabilizing PPARγ

Lihua Wang[1,5], Yuqin Wang[1,2,5], Kaixin Ding[1,5], Zhenzhi Li[1,5], Zhipeng Zhang [1], Xinzhi Li[3], Yue Song[1], Liwei Xie[4] & Zheng Chen [1 ✉]

## Abstract

**Brown adipose tissue (BAT) plays a vital role in non-shivering thermogenesis and energy metabolism and is influenced by factors like environmental temperature, ageing, and obesity. However, the molecular mechanisms behind BAT development and thermogenesis are not fully understood. Our study identifies the m⁶A reader protein YTHDC1 as a crucial regulator of postnatal interscapular BAT development and energy metabolism in mice. YTHDC1 directly interacts with PPARγ through its intrinsically disordered region (IDR), thus protecting PPARγ from binding the E3 ubiquitin ligase ARIH2, and preventing its ubiquitin-mediated proteasomal degradation. Specifically, the ARIH2 RING2 domain is essential for PPARγ degradation, while PPARγ's A/B domain is necessary for their interaction. Deletion of *Ythdc1* in BAT increases PPARγ degradation, impairing interscapular BAT development, thermogenesis, and overall energy expenditure. These findings reveal a novel mechanism by which YTHDC1 regulates BAT development and energy homeostasis independently of its m⁶A recognition function.**

**Keywords** YTHDC1; Intrinsically Disordered Region; Brown Adipose Tissue; Thermogenesis; PPARγ
**Subject Categories** Chromatin, Transcription & Genomics; Development; Signal Transduction

## Introduction

Interscapular brown adipose tissue (iBAT) is a specialized tissue critical for non-shivering thermogenesis and energy expenditure. It generates heat by harnessing the mitochondrial electrochemical gradient, primarily through the action of uncoupling protein 1 (UCP1) (Lowell and Spiegelman, 2000; Wu et al, 2012). In rodents, iBAT undergoes significant postnatal development, reaching full thermogenic capacity in response to relatively low environmental temperature (Harms et al, 2014; Mouroux et al, 1990; Obregón et al, 1996; Obregón et al, 1989; Wang et al, 2020; Xue et al, 2007). Impaired iBAT development can lead to cold intolerance and reduced energy expenditure (Wang et al, 2020), which may contribute to the development of obesity. The thermogenic capacity of iBAT is regulated by several factors, including environmental temperature, ageing, obesity and genetics (Cannon and Nedergaard, 2004; Inagaki et al, 2016). Thus, understanding the molecular mechanisms underlying iBAT development and thermogenesis is crucial for identifying novel therapeutic targets for obesity management.

PPARγ is a key transcription factor that regulates iBAT development by controlling essential genes such as *Prdm16*, *Ppargc1a*, and *Ucp1* (Harms et al, 2014; Harms et al, 2015; Inagaki et al, 2016; Lefterova and Lazar, 2009; Puigserver et al, 1998; Seale et al, 2007), which are critical for iBAT function. PPARγ is also important for the development of white adipose tissue (WAT). Mice lacking *Pparg* exhibit a complete absence of both BAT and WAT (Barak et al, 1999; Rosen et al, 1999), and targeted deletion of *Pparg* in whole adipocytes leads to decreased iBAT and WAT size (He et al, 2003; Jones et al, 2005; Wang et al, 2013). Brown adipocyte specific deletion of *Pparg* also results in reduced iBAT size (Xiong et al, 2018). PPARγ is regulated at both the transcriptional and post-transcriptional levels. RNA-binding proteins (RBPs) are major regulators of RNA processing and play critical roles in post-transcriptional regulation. Recent findings emphasize the importance of N6-methyladenosine (m⁶A) modification in regulating *Pparg* expression, which is crucial for BAT development and energy expenditure. Notably, deletion of m⁶A writer proteins such as METTL3 or WTAP in BAT decreases m⁶A modification and *Pparg* expression, impairing iBAT development and reducing energy expenditure (Wang et al, 2020; Wang et al, 2022b). The cytosolic m⁶A reader protein YTHDF2 has been implicated in recognizing m⁶A modifications on *Pparg* transcripts

---

[1]HIT Center for Life Sciences, School of Life Science and Technology, State Key Laboratory of Matter Behaviors in Space Environment, Frontier Science Center for Interaction between Space Environment and Matter, Zhengzhou Research Institute, Harbin Institute of Technology, Harbin 150001, China. [2]Department of Cardiovascular Surgery, Institute for Chronic Diseases, The Affiliated Hospital of Qingdao University, Qingdao 266000, China. [3]NHC Key Laboratory of Cell Transplantation, First Affiliated Hospital of Harbin Medical University, Harbin 150001, China. [4]Guangdong Provincial Key Laboratory of Microbial Culture Collection and Application, State Key Laboratory of Applied Microbiology Southern China, Institute of Microbiology, Guangdong Academy of Sciences, Guangzhou 510070, China. [5]These authors contributed equally: Lihua Wang, Yuqin Wang, Kaixin Ding, Zhenzhi Li. ✉E-mail: chenzheng@hit.edu.cn

 

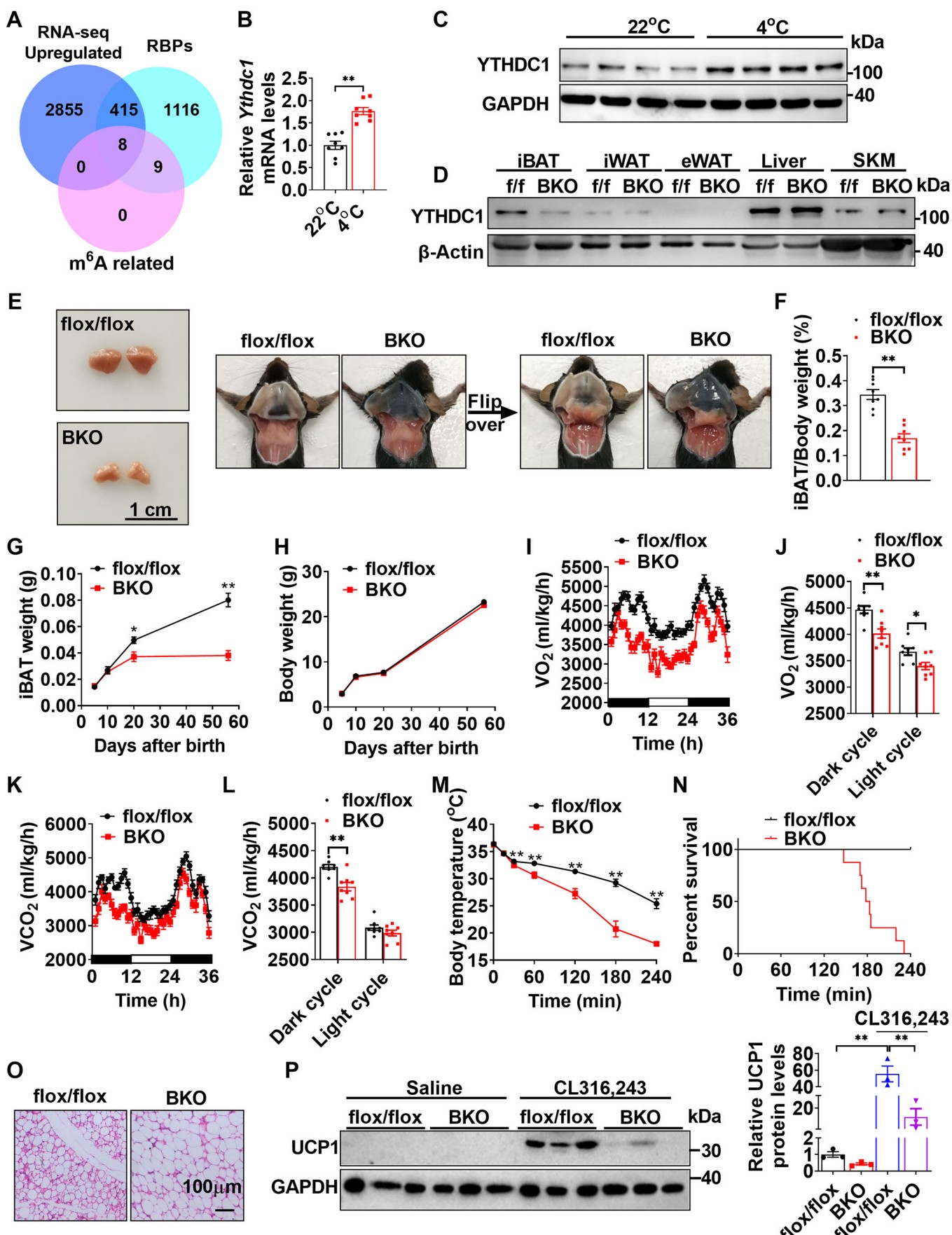

**◄ Figure 1. Identification of YTHDC1 as a key regulator of iBAT development, energy expenditure, and thermogenesis.**

(A) A Venn diagram from RNA-seq data (GEO dataset GSE234972) revealed that showed that 423 RBPs and 8 m⁶A related RBPs were upregulated during cold exposure. (B) *Ythdc1* mRNA levels in iBATs during cold exposure were measured by RT-qPCR ($n = 8$ per group; $P < 0.0001$). (C) YTHDC1 protein levels in iBATs during cold exposure were determined by immunoblotting ($n = 4$ per group). (D) YTHDC1 protein levels in iBAT, iWAT, eWAT, liver, and skeletal muscle of *Ythdc1*$^{flox/flox}$ and *Ythdc1*-BKO mice at 8 weeks of age. (E) Gross appearance of iBATs in *Ythdc1*$^{flox/flox}$ and *Ythdc1*-BKO mice at 8 weeks of age. (F) Relative iBAT weight of *Ythdc1*$^{flox/flox}$ and *Ythdc1*-BKO mice at 8 weeks old ($n = 8$ per group; $P < 0.0001$). (G) iBAT weight of *Ythdc1*$^{flox/flox}$ and *Ythdc1*-BKO mice at 5, 10, 20, and 56 days after birth ($n = 5$–8 for each group; 5 days: $P = 0.5120$; 10 days: $P = 0.6999$; 20 days: $P = 0.0175$; 56 days: $P = 0.0002$). (H) Body weight of *Ythdc1*$^{flox/flox}$ and *Ythdc1*-BKO mice at 5, 10, 20, and 56 days after birth ($n = 5$–8 for each group). (I, J) The O₂ consumption rates in 8-week-old *Ythdc1*$^{flox/flox}$ and *Ythdc1*-BKO mice at 22 °C ($n = 8$ per group; Dark cycle: $P = 0.0012$; Light cycle: $P = 0.0123$). (K, L) The CO₂ production rates in 8-week-old *Ythdc1*$^{flox/flox}$ and *Ythdc1*-BKO mice at 22 °C ($n = 8$ per group; Dark cycle: $P = 0.0008$; Light cycle: $P = 0.1800$). (M) The rectal temperature of 8-week-old *Ythdc1*$^{flox/flox}$ and *Ythdc1*-BKO mice during acute cold exposure (4 °C) ($n = 8$ per group; 0 min: $P = 0.3523$; 15 min: $P = 0.8013$; 30 min: $P = 0.0053$; 60 min: $P = 0.0071$; 120 min: $P = 0.0007$; 180 min: $P = 0.0011$; 240 min: $P = 0.0051$). (N) Percent survival of 8-week-old *Ythdc1*$^{flox/flox}$ and *Ythdc1*-BKO mice during acute cold exposure (4 °C) ($n = 8$ per group). (O, P) *Ythdc1*$^{flox/flox}$ and *Ythdc1*-BKO mice were injected with CL 316,243 for 4 days to induce browning of WAT. H&E staining of iWAT in *Ythdc1*$^{flox/flox}$ and *Ythdc1*-BKO mice after CL 316,243 treatment was shown (O). UCP1 protein levels in iWAT of *Ythdc1*$^{flox/flox}$ and *Ythdc1*-BKO mice treated with or without CL 316,243 were measured by immunoblotting (P) ($n = 3$ per group; flox/flox+CL316,243 VS flox/flox+Saline: $P = 0.000165$; flox/flox+CL316,243 VS BKO + CL316,243: $P = 0.000829$). *$P < 0.05$. **$P < 0.01$. Data represent the mean ± SEM. Differences between two groups were analyzed by unpaired two-tailed Student's $t$ tests. Differences among more than two groups were analyzed by one-factor analysis of variance (ANOVA). $n$ was the number of biologically independent mice. Source data are available online for this figure.

and influencing its expression (Wang et al, 2020). In addition to post-transcriptional regulation, PPARγ is also modulated at the post-translational level, particularly through ubiquitination by E3 ubiquitin ligases such as NEDD4 and SMURF1 (Li et al, 2016; Zhu et al, 2018). These ubiquitination processes regulate PPARγ protein abundance and transcriptional activity. However, the intricate regulatory mechanisms governing PPARγ at both post-transcriptional and post-translational levels remain largely unexplored.

Beyond cytosolic m⁶A reader proteins, nuclear m⁶A reader proteins such as YTH domain containing 1 (YTHDC1) and YTHDC2 also play significant roles in RNA regulation. YTHDC1, primarily located within nuclear speckles, binds to specific single-stranded RNA sequences and recognizes m⁶A modifications through its YTH domain (Nayler et al, 2000). This interaction is essential for regulating various RNA processing events, including RNA splicing, nuclear RNA export, and RNA decay (Liu et al, 2020; Roundtree et al, 2017; Shima et al, 2017; Xiao et al, 2016; Xu et al, 2014; Zhang et al, 2010). YTHDC1 also contains two large intrinsically disordered regions (IDRs), although the functional role of these regions remains unknown. Recent studies have highlighted additional roles for YTHDC1, including the regulation of chromatin accessibility (Liu et al, 2020; Woodcock et al, 2020). Deletion of *Ythdc1* leads to embryonic lethality, underscoring its critical role in tissue development (Kasowitz et al, 2018). However, the potential involvement of YTHDC1 in the iBAT development and thermogenesis has not yet been investigated.

In this study, we have demonstrated that YTHDC1 is crucial for the development and energy metabolism of brown adipose tissue by interacting with PPARγ through its intrinsically disordered region (IDR). This interaction prevents the binding of the E3 ubiquitin ligase ARIH2 to PPARγ, thereby protecting PPARγ from degradation. Conversely, knockout of *Ythdc1* in brown adipose tissue (*Ythdc1*-BKO) increases PPARγ's exposure to ARIH2, leading to its degradation. This results in impaired iBAT development, reduced iBAT size and thermogenesis, decreased energy expenditure, and exacerbated HFD-induced glucose intolerance and insulin resistance. Our findings reveal a novel molecular mechanism by which YTHDC1 regulates iBAT development and maintains energy balance.

## Results

### Identification of YTHDC1 as a key regulator in iBAT development, energy expenditure, and thermogenesis

To investigate which RNA-binding proteins (RBPs) regulate BAT function, we analyzed a previously published GEO dataset GSE234972 (Levy et al, 2023). As shown in Fig. 1A and Appendix Fig. S1A, cold exposure led to the upregulation of 423 RBPs and the downregulation of 161 RBPs, suggesting an important role for RBPs in thermogenesis. Gene ontology (GO) analysis revealed that these RBPs were associated with RNA processing, RNA metabolic process, translation, ribonucleoprotein complex biogenesis, ribosome biogenesis, RNA splicing, and regulation of RNA stability (Appendix Fig. S1B). Among the differentially regulated RBPs, we observed that eight m⁶A-related RBPs (YTHDC1, YTHDF1-3, METTL3, METTL14, RBM15, and HNRNPA2B1) were upregulated (Fig. 1A; Appendix Fig. S1C), while two m⁶A-related RBPs (FTO and FMR1) were downregulated (Appendix Fig. S1A). Interestingly, m⁶A writers (METTL3, METTL14, RBM15) showed upregulation, whereas m⁶A eraser FTO was downregulated. Consistently with these findings, our previous research demonstrated that both METTL3 and YTHDF2 were upregulated during cold exposure (Wang et al, 2020). We further confirmed that YTHDC1 mRNA and protein levels were significantly induced by cold exposure (4 °C for 6 h) (Fig. 1B,C). These data suggest that YTHDC1 may regulate iBAT function and thermogenesis.

To investigate whether YTHDC1 regulates iBAT function and thermogenesis, we created BAT-specific *Ythdc1* knockout (*Ythdc1*-BKO) mice by crossing *Ythdc1*-floxed mice (Appendix Fig. S2A,B) with *Ucp1*-iCre transgenic mice. In these transgenic mice, IRES-Cre was inserted between exon 6 and the 3′ untranslated region (UTR), enabling concurrent expression of *Ucp1* and iCre expression (Li et al, 2017). Since *Ucp1* expression begins after day 18 of gestation and rapidly increases until 10 days postnatally (Xue et al, 2007), this model was designed to delete genes in iBAT shortly after birth. In 8-week-old *Ythdc1*-BKO mice, YTHDC1 was successfully deleted in iBAT, but not in other tissues such as iWAT, liver, and skeletal muscle (Fig. 1D). We observed no significant differences in body weight (Appendix Fig. S2C), iBAT morphology (Appendix Fig. S2D), iBAT weight (Appendix

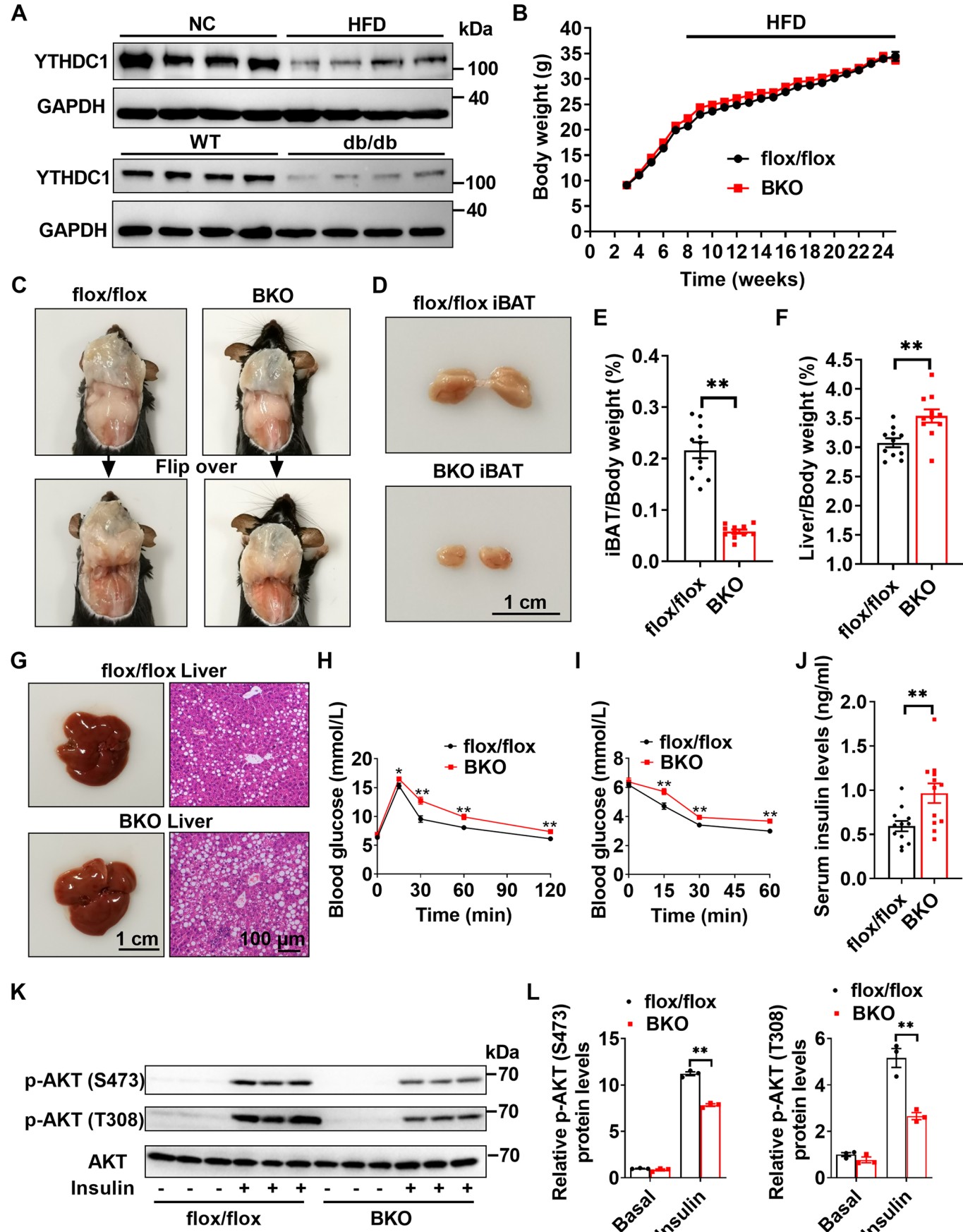

**Figure 2. Brown adipocyte-specific knockout of *Ythdc1* promotes HFD-induced glucose intolerance and insulin resistance.**

(A) The YTHDC1 and GAPDH protein levels in iBATs of 16-week HFD-fed mice, 8-week-old db/db mice, and their respective control mice were measured by immunoblotting ($n = 4$ per group). (B) The body weights of *Ythdc1*-BKO and *Ythdc1*[flox/flox] mice fed HFD diet for 17 weeks ($n = 12$ per group). (C, D) The morphology of iBATs in 25-week-old HFD-fed mice. (E) The relative iBAT weight in 25-week-old HFD-fed mice ($n = 11$ per group; $P < 0.0001$). (F) The relative liver weight in 25-week-old HFD-fed mice ($n = 11$ per group; $P = 0.003$). (G) The morphology of liver and H&E staining of liver sections in 25-week-old HFD-fed mice. (H) Glucose tolerance tests (GTTs) were conducted in 24-week-old HFD-fed *Ythdc1*[flox/flox] and *Ythdc1*-BKO mice ($n = 8$ per group; 0 min: $P = 0.1244$; 15 min: $P = 0.0359$; 30 min: $P = 0.0016$; 60 min: $P = 0.0037$; 120 min: $P = 0.0023$). (I) Insulin tolerance tests (ITTs) were performed in 24-week-old HFD-fed *Ythdc1*[flox/flox] and *Ythdc1*-BKO mice ($n = 8$ per group; 0 min: $P = 0.8149$; 15 min: $P = 0.0008$; 30 min: $P = 0.0098$; 60 min: $P = 0.0053$). (J) The serum insulin levels of 25-week-old HFD-fed *Ythdc1*[flox/flox] and *Ythdc1*-BKO mice ($n = 12$ per group; $P = 0.0072$). (K, L) Hepatic p-AKT (S473), p-AKT (T308), and AKT protein levels were measured by immunoblotting and quantified using ImageJ in 25-week-old HFD-fed *Ythdc1*[flox/flox] and *Ythdc1*-BKO mice ($n = 3$ per group; p-AKT (S473) BKO VS flox/flox: $P < 0.0001$; p-AKT (T308) BKO VS flox/flox $P < 0.0001$). Data represent the mean ± SEM. Significance was determined by unpaired two-tailed Student's *t* test analysis. Differences among more than two groups were analyzed by one-factor analysis of variance (ANOVA). *$P < 0.05$. **$P < 0.01$. *n* was the number of biologically independent mice. Source data are available online for this figure.

Fig. S2E), or cold challenge (Appendix Fig. S2F) between *Ythdc1*[flox/flox] and *Ucp1*-iCre mice. Thus, we used *Ythdc1*[flox/flox] mice as the control for subsequent experiments.

Interestingly, *Ythdc1*-BKO mice exhibited abnormal iBAT morphology, with visibly smaller and "whitened" iBAT (Fig. 1E,F). H&E and Oil Red O staining showed higher lipid accumulation in *Ythdc1*-BKO iBATs (Appendix Fig. S3A,B). Both immunoblotting and immunostaining showed that UCP1 expression was significantly decreased in *Ythdc1*-BKO iBATs (Appendix Fig. S3C,D). To determine when this phenotype developed, we isolated iBATs at different postnatal stages. By day 20 post-birth, *Ythdc1*-BKO mice already displayed smaller iBAT compared to *Ythdc1*[flox/flox] mice (Fig. 1G), although this did not affect overall body weight gain (Fig. 1H). We did not detect any TUNEL-positive cells or cleaved caspase 3 in both *Ythdc1*[flox/flox] and *Ythdc1*-BKO iBATs, indicating that *Ythdc1*-BKO does not induce cell apoptosis in iBATs. BAT-specific *Ythdc1* knockout did not affect iWAT or eWAT (Appendix Fig. S3E). These results suggest that BAT YTHDC1 is critical for the postnatal development of iBAT.

*Ythdc1*-BKO mice displayed lower oxygen ($O_2$) consumption and carbon dioxide ($CO_2$) production during both light and dark cycles (Fig. 1I–L), and increased physical activity (Appendix Fig. S3F), despite similar food intake compared to *Ythdc1*[flox/flox] mice (Appendix Fig. S3G), indicating a decrease in energy expenditure. Additionally, following cold exposure (4 °C), *Ythdc1*-BKO mice exhibited a rapid decline in body temperature, leading to death within four hours, whereas all *Ythdc1*[flox/flox] mice maintained their body temperature and survived (Fig. 1M,N). These data demonstrate that YTHDC1 in BAT is essential for thermogenesis.

We further investigated whether YTHDC1 regulates the browning of iWAT in response to β-adrenergic stimulation. *Ythdc1*-BKO and *Ythdc1*[flox/flox] mice were injected with the β-adrenergic agonist CL316,243 for 4 days to induce browning of WAT. As shown in Fig. 1O,P, CL316,243 treatment significantly induced browning of iWAT and increased UCP1 expression in *Ythdc1*[flox/flox] mice, but these effects were markedly attenuated in *Ythdc1*-BKO mice. These data suggest that BAT-specific deletion of *Ythdc1* impairs the browning of WAT in response to β-adrenergic stimulation, highlighting the critical role of YTHDC1 in energy expenditure regulation.

## BAT-specific knockout of *Ythdc1* promotes high-fat diet-induced glucose intolerance and insulin resistance

Reduced BAT thermogenesis is a known contributor to obesity in both rodents and humans (Tokuyama and Himms-Hagen, 1986; Vijgen et al,

2011). To assess whether *Ythdc1* expression in iBAT is associated with obesity, we examined YTHDC1 protein levels in two mouse models of obesity: high-fat diet (HFD)-induced obesity and leptin receptor-deficient db/db mice. YTHDC1 protein levels were significantly lower in both diet-induced obese (DIO) and db/db mice compared to controls (Fig. 2A), suggesting that YTHDC1 in iBAT may play a role in regulating energy metabolism and metabolic syndrome.

To determine whether *Ythdc1*-BKO mice are more susceptible to HFD-induced obesity, we fed *Ythdc1*-BKO and *Ythdc1*[flox/flox] mice with an HFD and monitored their body weights weekly. As shown in Fig. 2B, *Ythdc1*-BKO mice gained similar body weight to *Ythdc1*[flox/flox] mice during HFD feeding, indicating that *Ythdc1*-BKO mice are not more susceptible to HFD-induced obesity. One possible reason is the increased physical activity in *Ythdc1*-BKO mice (Appendix Fig. S3F). However, the morphology of iBATs in *Ythdc1*-BKO mice fed with HFD was abnormal, appearing smaller and showing signs of "whitening" (Fig. 2C,D). The relative weight of iBATs was significantly reduced in *Ythdc1*-BKO mice (Fig. 2E). H&E and Oil Red O staining showed higher lipid accumulation in *Ythdc1*-BKO iBATs (Appendix Fig. S3H,I). Both immunoblotting and immunostaining showed that UCP1 expression was significantly decreased in *Ythdc1*-BKO iBATs (Appendix Fig. S3J,K). In contrast, the relative liver weight was markedly increased in these mice (Fig. 2F), likely due to elevated triglyceride accumulation in the *Ythdc1*-BKO livers (Fig. 2G). Interestingly, the weights of iWAT and eWAT were similar between *Ythdc1*[flox/flox] and *Ythdc1*-BKO mice (Appendix Fig. S3L).

Next, we assessed whether YTHDC1 in BAT regulates systemic glucose homeostasis and insulin sensitivity. Mice were fed an HFD for 16 weeks (up to 24 weeks of age), followed by glucose tolerance tests (GTT) and insulin tolerance tests (ITT). *Ythdc1*-BKO mice exhibited significant glucose intolerance and insulin resistance compared to *Ythdc1*[flox/flox] mice (Fig. 2H,I). Consistent with these findings, serum insulin levels were elevated in *Ythdc1*-BKO mice (Fig. 2J), further confirming the development of insulin resistance. Additionally, insulin-induced phosphorylation of AKT at both S473 and T308 was significantly reduced in the livers of *Ythdc1*-BKO mice (Fig. 2K,L), indicating impaired insulin signaling. These results suggest that the loss of *Ythdc1* in BAT disrupts systemic energy homeostasis and promotes HFD-induced metabolic dysfunction, including glucose intolerance and insulin resistance.

## BAT-specific deletion of *Ythdc1* results in dramatically decreased expression of BAT-selective genes

To further investigate the molecular mechanisms underlying the impaired development of iBAT in *Ythdc1*-BKO mice, we

performed RNA sequencing (RNA-seq) analysis to examine the transcriptional profiles of iBATs from both *Ythdc1*-BKO and *Ythdc1*<sup>flox/flox</sup> mice. As shown in Fig. 3A and Dataset EV1, a total of 1643 genes were downregulated and 1550 genes were upregulated in *Ythdc1*-BKO iBATs. Gene Ontology (GO) analysis revealed that the downregulated genes were primarily involved in processes such as the generation of precursor metabolites and energy, cellular respiration, cofactor metabolism, purine ribonucleoside triphosphate metabolism, and ATP metabolic processes (Fig. 3B). In contrast, the upregulated genes were associated with immune response activation, defense responses to pathogens, phagocytosis, regulation of leukocyte activation, and cytokine production (Fig. 3C). KEGG pathway analysis further demonstrated that the downregulated genes were enriched in pathways related to oxidative phosphorylation, thermogenesis, carbon metabolism, propanoate metabolism, and the TCA cycle (Fig. 3D), while the upregulated genes were linked to pathways involving phagosomes, lysosomes, natural killer cell-mediated cytotoxicity, NOD-like receptor signaling, and antigen processing and presentation (Fig. 3E).

At the protein level, we observed a marked reduction in the expression of key thermogenic regulators such as UCP1, PPARγ, and PRDM16 in the iBATs of *Ythdc1*-BKO mice (Fig. 3F). Additionally, mitochondrial oxidative phosphorylation (OXPHOS) components, including those of complexes I, II, III, IV, and V, were dramatically reduced in *Ythdc1*-BKO iBATs (Fig. 3G). These findings indicate that BAT-specific deletion of *Ythdc1* impairs the development of iBAT by downregulating thermogenesis-related genes and mitochondrial oxidative phosphorylation pathways, thereby compromising the metabolic function of BAT.

## Fat-specific deletion of *Ythdc1* results in decreased iBAT/WAT mass and PPARγ protein levels

PPARγ is important for the development of both BAT and WAT. Targeted deletion of *Pparg* in whole adipocytes leads to decreased iBAT and WAT size (He et al, 2003; Jones et al, 2005; Wang et al, 2013). We asked whether YTHDC1 also regulates WAT development by affecting PPARγ expression. To answer this question, we generated fat-specific *Ythdc1* knockout (*Ythdc1*-FKO) mice by crossing *Ythdc1*<sup>flox/flox</sup> mice with *Adipoq*-Cre transgenic mice. As shown in Fig. 4A, YTHDC1 was specifically deleted in iBAT, iWAT, and eWAT, but not in the liver. *Ythdc1*-FKO significantly decreased weights of iBAT, iWAT, and eWAT (Fig. 4B), which is likely due to decreased PPARγ protein levels (Fig. 4C–E). Some PPARγ targeted genes such as *Fabp4*, *Plin2*, *Cd36*, *Glut4*, and *Adipoq* were significantly downregulated in both iWAT and eWAT of *Ythdc1*-FKO mice (Fig. 4F), which further leads to smaller WAT (Fig. 4B). *Ythdc1*-FKO mice also showed impaired thermogenesis after cold exposure (Fig. 4G), which was likely attributed to the downregulated UCP1 levels (Fig. 4C). Both *Ythdc1*-BKO and *Ythdc1*-FKO increased the expression of inflammation-related genes in iBATs (Fig. 4H,I). To further test whether deletion of *Ythdc1* in iBAT causes immune cell infiltration, we performed fluorescence activated cell sorting (FACS) analysis on stromal-vascular fraction (SVF) cells isolated from the iBATs of *Ythdc1*<sup>flox/flox</sup> and *Ythdc1*-FKO mice. As shown in Fig. 4J–L and Appendix Fig. S4A–C, total immune (CD45.2<sup>+</sup>) cells, macrophage (CD45.2<sup>+</sup> F4/80<sup>+</sup>), total T (CD45.2<sup>+</sup> CD3<sup>+</sup>) cells, CD4+ T (CD45.2<sup>+</sup> CD3<sup>+</sup> CD4<sup>+</sup>) cells, NK (CD45.2<sup>+</sup> NK1.1<sup>+</sup> CD3<sup>-</sup>) cells, B (CD45.2<sup>+</sup>B220<sup>+</sup>CD11c<sup>-</sup>) cells were significantly increased in iBATs of

*Ythdc1*-FKO mice, while dendritic cells (CD45.2<sup>+</sup>CD11c<sup>+</sup>CD11b<sup>-</sup>) and CD8+ T (CD45.2<sup>+</sup> CD3<sup>+</sup> CD8<sup>+</sup>) cells were not changed in iBATs of *Ythdc1*-FKO mice. These data demonstrate that the deletion of *Ythdc1* causes immune cell infiltration in iBAT, leading to higher expression of inflammation-related genes. These results suggest that YTHDC1 is important for the development of both BAT and WAT through the involvement of PPARγ.

## Reduced PPARγ in iBATs of *Ythdc1*-BKO mice contributes to decreased iBAT mass and impaired thermogenesis

We observed a significant reduction in PPARγ protein levels in the iBATs of *Ythdc1*-BKO mice, which became evident starting at twenty days of age (Fig. 5A). Given that *Pparg* knockout in either adipose tissue or brown adipose tissue results in smaller iBAT mass (He et al, 2003; Jones et al, 2005; Wang et al, 2013; Xiong et al, 2018), similar to the phenotype seen in *Ythdc1*-BKO mice, it suggests that the decreased PPARγ expression may contribute to the reduced iBAT mass in these mice. To test this hypothesis, we re-expressed PPARγ in the iBATs of *Ythdc1*-BKO mice via local injection of a purified PPARγ adenovirus. As shown in Fig. 5B–F, re-expression of PPARγ in *Ythdc1*-BKO iBATs restored iBAT weight, improved thermogenic capacity, and increased the expression of UCP1 and mitochondrial OXPHOS proteins. These findings indicate that the reduced PPARγ levels in *Ythdc1*-BKO iBATs play a critical role in the observed reductions in iBAT mass and thermogenesis.

## YTHDC1 regulates PPARγ expression, iBAT development, and thermogenesis through its intrinsically disordered region (IDR)

AlphaFold2 (Jumper et al, 2021), ANCHOR2 (Dosztányi et al, 2009), and IUPred2 (Mészáros et al, 2018) predicted that YTHDC1 contains two large intrinsically disordered regions (IDRs) and a YTH domain (Fig. 6A). The YTH domain is essential for recognizing m<sup>6</sup>A modified RNA and regulating RNA processing (Xu et al, 2014). Specifically, the W378 residue in the YTH domain is crucial for m<sup>6</sup>A recognition, as shown by studies in which the YTHDC1 W378A mutation disrupts m<sup>6</sup>A binding (Xu et al, 2014). To determine whether YTHDC1's role in promoting iBAT development depends on m<sup>6</sup>A recognition or its IDR, we evaluated whether re-expressing YTHDC1, YTHDC1 W378A (which cannot bind m<sup>6</sup>A), or YTHDC1 ΔIDR (lacking the IDR) in the iBATs of six-week-old *Ythdc1*-BKO mice would rescue the phenotypes. We delivered adenoviruses expressing either wild-type YTHDC1 (Ad-YTHDC1), YTHDC1 W378A (Ad-W378A), YTHDC1 ΔIDR (Ad-ΔIDR), or a control β-galactosidase (Ad-βGal) directly into the iBATs. Re-expression of wild-type YTHDC1 or the m<sup>6</sup>A-binding-deficient YTHDC1 W378A, but not YTHDC1 ΔIDR, significantly restored iBAT function in *Ythdc1*-BKO mice. This was demonstrated by increased iBAT weight (Fig. 6B), enhanced thermogenesis (Fig. 6C), and elevated expression of key thermogenic regulators such as UCP1, PPARγ, PGC-1α, PRDM16, and mitochondrial OXPHOS components (complexes II, III, IV, and V) (Fig. 6D,E). These data indicate that the intrinsically disordered region (IDR) of YTHDC1, rather than its m<sup>6</sup>A recognition capacity, is essential for the development of iBAT and its thermogenic function.

## A

**BKO VS flox/flox**

• DOWN 1643  • UP 1550

-log10(padj) vs log2FoldChange

## B

GO analysis of downregulated genes

| Description | Count | pvalue |
|---|---|---|
| generation of precursor metabolites and energy | 135 | 3.74E-56 |
| cellular respiration | 89 | 3.40E-53 |
| cofactor metabolic process | 152 | 5.33E-50 |
| purine ribonucleoside triphosphate metabolic process | 104 | 2.14E-46 |
| ATP metabolic process | 97 | 4.26E-45 |

## C

GO analysis of upregulated genes

| Description | Count | pvalue |
|---|---|---|
| positive regulation of immune response | 116 | 2.28E-25 |
| defense response to other organism | 96 | 3.04E-22 |
| phagocytosis | 58 | 2.55E-20 |
| regulation of leukocyte activation | 102 | 4.59E-20 |
| positive regulation of cytokine production | 86 | 8.17E-18 |

## D

KEGG analysis of downregulated genes

| KEGGID | Description | Count | pvalue |
|---|---|---|---|
| mmu00190 | Oxidative phosphorylation | 79 | 9.60E-44 |
| mmu04714 | Thermogenesis | 96 | 2.65E-34 |
| mmu01200 | Carbon metabolism | 59 | 4.75E-27 |
| mmu00640 | Propanoate metabolism | 27 | 9.87E-24 |
| mmu00020 | Citrate cycle (TCA cycle) | 24 | 6.55E-18 |

## E

KEGG analysis of upregulated genes

| KEGGID | Description | Count | pvalue |
|---|---|---|---|
| mmu04145 | Phagosome | 45 | 5.28E-10 |
| mmu04142 | Lysosome | 33 | 4.15E-07 |
| mmu04650 | Natural killer cell mediated cytotoxicity | 27 | 4.18E-06 |
| mmu04621 | NOD-like receptor signaling pathway | 35 | 1.72E-05 |
| mmu04612 | Antigen processing and presentation | 21 | 2.11E-05 |

## F

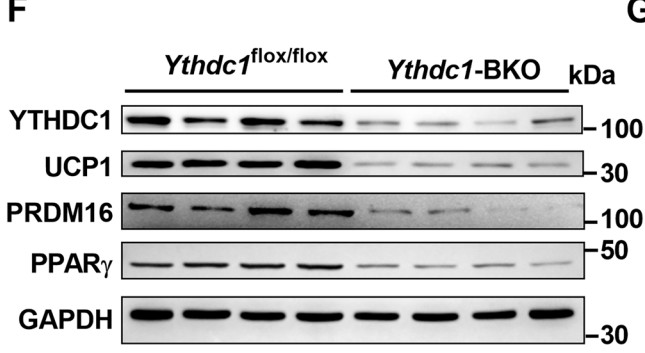

## G

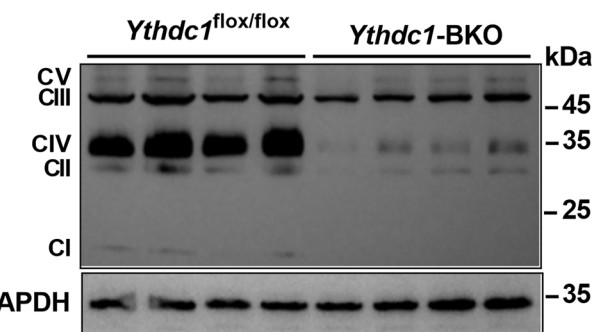

## YTHDC1 regulates PPARγ protein stability through its IDR involving the E3 ligase ARIH2

*Ythdc1*-BKO caused significantly decreased PPARγ protein levels in the iBATs, and *Ythdc1*-FKO led to a significant reduction of PPARγ protein levels in both BAT and WAT (Figs. 3F and 4C–E). However, *Pparg* mRNA levels remained unchanged (Appendix Fig. S5A–D). These data suggest that *Ythdc1* deficiency impacts PPARγ protein stability. To investigate this, we performed protein stability assays and found that deletion of *Ythdc1* in primary brown

◀ **Figure 3.  Brown adipocyte-specific deletion of *Ythdc1* dramatically decreases expression of BAT-selective genes.**

RNA-seq analysis was performed in the iBATs of *Ythdc1*^flox/flox and *Ythdc1*-BKO mice at 8 weeks old (*n* = 3 per group). (**A**) The differentially expressed genes (DEGs) (*Ythdc1*-BKO VS flox/flox; *n* = 3 per group) including 1643 downregulated genes and 1550 upregulated genes were illustrated in a volcano plot (|log2FoldChange| >0 and padj < 0.05). Differential expression analysis was performed using the DESeq2 R package (1.20.0). DESeq2 provided statistical routines for determining differential expression in digital gene expression data using a model based on the negative binomial distribution. The resulting *P*-values were adjusted using the Benjamini and Hochberg's approach for controlling the false discovery rate. (**B, C**) GO biological process terms enriched in downregulated and upregulated genes. (**D, E**) KEGG analysis of downregulated and upregulated genes. (**F**) UCP1, PPARγ, PRDM16, and GAPDH protein levels in iBATs of 8-week-old *Ythdc1*^flox/flox and *Ythdc1*-BKO mice were determined by immunoblotting (*n* = 4 per group). (**G**) Mitochondrial complex protein levels in iBATs of 8-week-old *Ythdc1*^flox/flox and *Ythdc1*-BKO mice were determined by immunoblotting (*n* = 4 per group). *n* was the number of biologically independent mice. Source data are available online for this figure.

adipocytes markedly decreased PPARγ protein stability (Fig. 7A). Treatment with MG132, a proteasome inhibitor, resulted in increased PPARγ protein levels in the iBATs of *Ythdc1*-BKO mice (Fig. 7B), while leupeptin, a lysosomal inhibitor, did not increase PPARγ levels (Fig. 7C), indicating that the degradation of PPARγ is primarily dependent on the proteasome.

We then examined whether YTHDC1 directly interacts with PPARγ to stabilize it. Co-immunoprecipitation assays showed that YTHDC1 interacted with PPARγ in iBATs (Appendix Fig. S6A). Co-immunoprecipitation assays revealed that both wild-type YTHDC1 and its W378A mutation, but not the YTHDC1 ΔIDR variant, interacted with PPARγ in HEK293T cells (Fig. 7D), primary brown adipocytes (Fig. 7E), and primary white adipocytes (Appendix Fig. S6B). Furthermore, both YTHDC1 and its W378A mutation, but not the YTHDC1 ΔIDR, were able to reverse the downregulation of PPARγ protein levels observed in *Ythdc1*-BKO iBATs (Fig. 6D,E), demonstrating that the IDR is essential for YTHDC1-mediated stabilization of PPARγ, independent of m⁶A recognition.

To identify the specific E3 ligase responsible for PPARγ degradation in the absence of YTHDC1, we employed UbiBrowser2.0 (Wang et al, 2022a) to predict E3 ubiquitin ligases that could target PPARγ. From this analysis, thirty-two E3 ligases were identified as potential candidates (Appendix Fig. S7A), with fifteen of these localized to the nucleus and expressed in iBATs (Appendix Fig. S7B). We first screened these nuclear E3 ligases and found that ARIH2 was capable of degrading PPARγ (Fig. 7F). This degradation occurred in a dose-dependent manner (Fig. 7G). Further analysis revealed that both wild-type ARIH2 and its H158A mutant, but not the C300A mutant, promoted PPARγ degradation (Fig. 7H,I), suggesting that the C300 residue in the RING2 domain, but not H158 in the RING1 domain, is critical for ARIH2-mediated degradation of PPARγ.

Co-immunoprecipitation experiments confirmed that ARIH2 directly binds to PPARγ in both HEK293T cells and primary brown adipocytes (Fig. 7J,K). However, this interaction was detectable only when cells were treated with the proteasome inhibitor MG132, indicating that ARIH2 rapidly degrades PPARγ under normal conditions (Fig. 7J,K). Further domain mapping showed that the A/B domain of PPARγ, but not other domains, was responsible for its interaction with ARIH2 (Fig. 7L,M). Consistently, ARIH2 was unable to degrade a PPARγ mutant lacking the A/B domain (PPARγΔAB), highlighting the importance of this region for ARIH2-mediated degradation (Fig. 7N). Additionally, ARIH2 promoted the ubiquitination of PPARγ (Fig. 7O; Appendix Fig. S7C), leading to its proteasome-dependent degradation.

Next, we explored how YTHDC1 and its intrinsically disordered region (IDR) protect PPARγ from ARIH2-mediated degradation. Co-transfection and co-infection experiments showed that both YTHDC1 and its W378A mutant, but not the YTHDC1 ΔIDR variant, were able to prevent ARIH2-induced degradation of PPARγ in HEK293T cells (Fig. 8A), primary brown adipocytes (Fig. 8B), and primary white adipocytes (Appendix Fig. S8A). Interestingly, YTHDC1 formed nuclear puncta with PPARγ in an IDR-dependent manner in HEK293T cells (Fig. 8C; Appendix Fig. S8B), primary brown adipocytes (Fig. 8D; Appendix Fig. S8C), and primary white adipocytes (Appendix Fig. S8D,E), suggesting that YTHDC1, through its IDR, physically interacts with PPARγ to shield it from ARIH2 binding and subsequent degradation. In *Ythdc1*-BKO mice, the absence of YTHDC1 increases PPARγ exposure to ARIH2, leading to enhanced degradation of PPARγ.

Finally, we investigated whether ARIH2 is required for PPARγ degradation in the iBATs of *Ythdc1*-BKO mice. We performed *Arih2* knockdown experiments by injecting purified Ad-shArih2-1 or Ad-shArih2-2 adenoviruses into the iBATs of *Ythdc1*-BKO mice. Knockdown of *Arih2* by two individual Ad-shRNAs (Ad-shArih2-1 and Ad-shArih2-2) significantly rescued iBAT development, as evidenced by increased iBAT mass (Fig. 8E), enhanced thermogenesis (Fig. 8F), and upregulated expression of BAT-related genes, which correlated with elevated PPARγ protein levels (Fig. 8G,H). These findings demonstrate that *Ythdc1* deficiency impairs iBAT development and thermogenesis by promoting ARIH2-mediated degradation of PPARγ.

## Discussion

RNA-binding proteins (RBPs) play crucial roles in RNA processing, influencing numerous physiological and pathological processes. Many RBPs contain intrinsically disordered regions (IDRs), which are vital for protein aggregation and liquid-liquid phase separation (Lin et al, 2015). However, the physiological roles of IDRs in RBPs remain largely unexplored. In this study, we demonstrated that YTHDC1 regulates development of iBAT and thermogenesis by preventing ARIH2-mediated degradation of PPARγ through its IDRs.

Notably, both YTHDC1 and m⁶A writer proteins (METTL3 and WTAP) are upregulated in response to cold stimulation in iBATs. Knockout models for *Ythdc1*, *Mettl3*, or *Wtap* in BAT reveal impaired postnatal iBAT development and thermogenesis, with all three models exhibiting iBAT whitening. Interestingly, *Ythdc1*-BKO mice display smaller iBAT, while *Mettl3*-BKO and *Wtap*-BKO mice present larger iBAT. All three mouse models show decreased

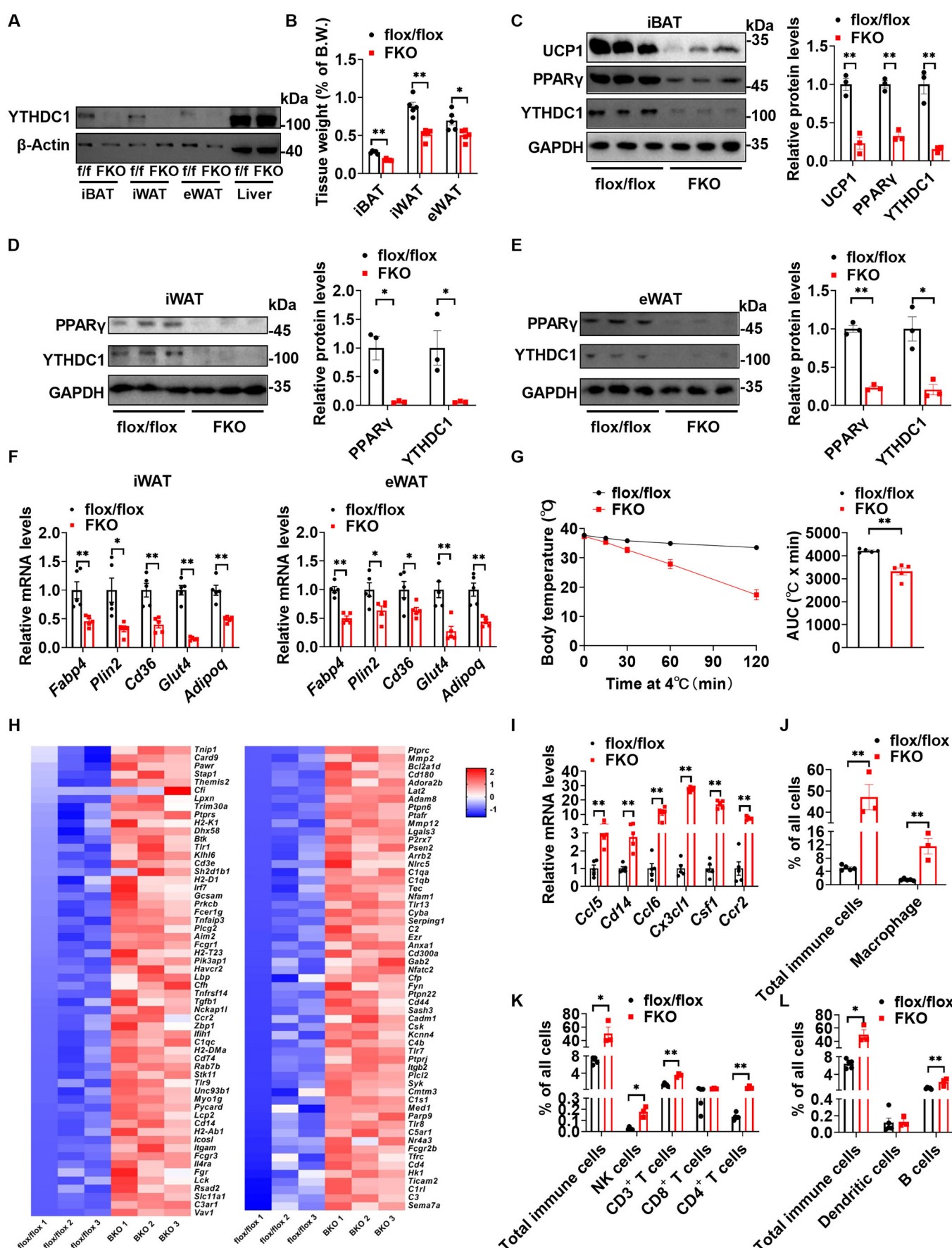

◀

**Figure 4.  Fat-specific deletion of *Ythdc1* decreases both white and brown fat.**

(A) YTHDC1 and β-Actin protein levels in iBAT, iWAT, eWAT, and liver of *Ythdc1*<sup>flox/flox</sup> and *Ythdc1*-FKO mice at 8 weeks of age were measured by immunoblotting. (B) Relative weight of iBAT, iWAT, and eWAT in *Ythdc1*<sup>flox/flox</sup> and *Ythdc1*-FKO mice at 8 weeks of age were measured ($n = 5$ per group; iBAT: $P = 0.0079$; iWAT: $P = 0.0079$; eWAT: $P = 0.0192$). (C) UCP1, PPARγ, YTHDC1, and GAPDH protein levels in iBATs of 8-week-old *Ythdc1*<sup>flox/flox</sup> and *Ythdc1*-FKO mice were determined by immunoblotting ($n = 3$ per group; UCP1: $P = 0.0023$; PPARγ: $P = 0.0004$; YTHDC1: $P = 0.0026$). (D) PPARγ, YTHDC1, and GAPDH protein levels in iWATs of 8-week-old *Ythdc1*<sup>flox/flox</sup> and *Ythdc1*-FKO mice were determined by immunoblotting ($n = 3$ per group; PPARγ: $P = 0.0103$; YTHDC1: $P = 0.0351$). (E) PPARγ, YTHDC1, and GAPDH protein levels in eWATs of 8-week-old *Ythdc1*<sup>flox/flox</sup> and *Ythdc1*-FKO mice were determined by immunoblotting ($n = 3$ per group; PPARγ: $P = 0.0001$; YTHDC1: $P = 0.0101$). (F) The *Fabp4*, *Plin2*, *Cd36*, *Glut4*, and *Adipoq* mRNA levels in iWAT and eWAT of 8-week-old *Ythdc1*<sup>flox/flox</sup> and *Ythdc1*-FKO mice were measured by RT-qPCR ($n = 5$ per group; In iWAT, FabP4: $P = 0.0077$; Plin2: $P = 0.0144$; Cd36: $P = 0.0019$; Glut4: $P < 0.0001$; Adipoq: $P = 0.0005$; In eWAT, FabP4: $P < 0.0001$; Plin2: $P = 0.0346$; Cd36: $P = 0.0439$; Glut4: $P = 0.0079$; Adipoq: $P = 0.0017$). (G) The rectal temperature of 8-week-old *Ythdc1*<sup>flox/flox</sup> and *Ythdc1*-FKO mice during acute cold exposure (4 °C) ($n = 5$ per group; $P = 0.0003$). (H) Heatmap shows that inflammatory genes were upregulated in iBATs of *Ythdc1*-BKO mice. (I) The *Ccl5*, *Cd14*, *Ccl6*, *Cx3cl1*, *Csf1*, and *Ccr2* mRNA levels in iBATs of *Ythdc1*-FKO mice were measure by RT-qPCR ($n = 5$ per group; Ccl5: $P = 0.0079$; Cd14: $P = 0.0005$; Ccl6: $P = 0.0006$; Cx3cl1: $P < 0.0001$; Csf1: $P < 0.0001$; Ccr2: $P < 0.0001$). (J–L) SVF cells were isolated from iBATs of *Ythdc1*<sup>flox/flox</sup> and *Ythdc1*-FKO mice, and fluorescence activated cell sorting (FACS) analysis were performed. Total immune (CD45.2<sup>+</sup>) cells, macrophage (CD45.2<sup>+</sup> F4/80<sup>+</sup>), total T (CD45.2<sup>+</sup> CD3<sup>+</sup>) cells, CD4 + T (CD45.2<sup>+</sup> CD3<sup>+</sup> CD4<sup>+</sup>) cells, CD8 + T (CD45.2<sup>+</sup> CD3<sup>+</sup> CD8<sup>+</sup>) cells, NK (CD45.2<sup>+</sup> NK1.1<sup>+</sup> CD3<sup>-</sup>) cells, dendritic (CD45.2<sup>+</sup>CD11c<sup>+</sup>CD11b<sup>-</sup>) cells, and B (CD45.2<sup>+</sup>B220<sup>+</sup>CD11c<sup>-</sup>) cells in iBATs of *Ythdc1*<sup>flox/flox</sup> and *Ythdc1*-FKO mice were quantified ($n = 3$–5 for each group; In (J), Total immune cells: $P < 0.0001$; Macrophage: $P = 0.0011$; In (K), Total immune cells: $P = 0.0357$; NK cells: $P = 0.0357$; CD3 + T cells: $P = 0.0001$; CD8 + T cells: $P = 0.1640$; CD4 + T cells: $P = 0.0065$; In (L), Total immune cells: $P = 0.0357$; Dendritic cells: $P = 0.5714$; B cells: $P = 0.0091$). Data represent the mean ± SEM. Significance was determined by unpaired two-tailed Student's *t* test analysis. *$P < 0.05$. **$P < 0.01$. *n* was the number of biologically independent mice. Source data are available online for this figure.

energy expenditure. Only *Mettl3*-BKO promotes HFD-induced obesity. Neither *Wtap*-BKO nor *Ythdc1*-BKO mice promote HFD-induced obesity, which is likely attributed to the increased physical activity observed in both *Wtap*-BKO and *Ythdc1*-BKO mice. It is possible that the deletion of *Wtap* or *Ythdc1* specifically in brown adipose tissue alters the expression and secretion of certain adipokines, which may then influence skeletal muscle or brain activity to regulate physical movement. Further investigation is needed to explore this hypothesis.

*Ythdc1*-BKO significantly reduces PPARγ protein levels, which is critical for the development of both BAT and white adipose tissue. *Pparg*-deficient mice lack both brown and white adipose tissues (Barak et al, 1999; Rosen et al, 1999), and specific *Pparg* knockout in either adipose tissue or brown adipose tissue results in smaller iBAT (He et al, 2003; Jones et al, 2005; Wang et al, 2013; Xiong et al, 2018), resembling the phenotype observed in *Ythdc1*-BKO mice. Restoration of PPARγ expression in *Ythdc1*-BKO iBATs rescues impaired iBAT development and thermogenesis, with significant upregulation of PPARγ targets such as PRDM16 and UCP1 following re-expression. These findings underscore the importance of PPARγ in mediating the effects of YTHDC1 and suggest that reduced PPARγ levels in *Ythdc1*-BKO mice contribute to the observed thermogenic defects and altered adipose tissue development.

Our study demonstrates that *Ythdc1* deficiency in either fat or BAT leads to a significant reduction in PPARγ protein levels, without affecting its mRNA levels. This suggests that YTHDC1 regulates PPARγ at the post-translational level. Interestingly, the m<sup>6</sup>A recognition mutant of YTHDC1 (W378A) was able to rescue the reduction in PPARγ protein, indicating that m<sup>6</sup>A recognition by YTHDC1 is not required for the regulation of PPARγ protein stability. Instead, YTHDC1 interacts directly with PPARγ, and this interaction is crucial for maintaining PPARγ protein stability. The intrinsically disordered region (IDR) of YTHDC1 plays a key role in this interaction, as YTHDC1 forms nuclear puncta with PPARγ in an IDR-dependent manner. Although previous studies have shown that the m<sup>6</sup>A recognition function of YTHDC1 influences phase separation and dot formation (Cheng et al, 2021), our findings indicate that m<sup>6</sup>A recognition is not involved in YTHDC1-

mediated stabilization of PPARγ protein. Rather, YTHDC1 prevents PPARγ degradation by shielding it from proteasome-mediated degradation, which is likely mediated by the E3 ubiquitin ligase ARIH2. Both YTHDC1 and its W378A mutant, but not the IDR-deleted variant (YTHDC1 ΔIDR), protected PPARγ from ARIH2-induced degradation. This highlights the critical role of the IDR in YTHDC1's protective function. In *Ythdc1*-BKO mice, the absence of YTHDC1 increases the exposure of PPARγ to ARIH2, resulting in its enhanced degradation. Notably, knockdown of *Arih2* in the iBATs of *Ythdc1*-BKO mice significantly promoted iBAT postnatal development by restoring PPARγ protein levels. These findings suggest that ARIH2-mediated degradation of PPARγ contributes to the impaired iBAT development and thermogenesis observed in *Ythdc1*-BKO mice. Further investigation into the role of ARIH2 in regulating both brown and white adipose tissues by targeting PPARγ is warranted. Understanding the broader implications of ARIH2-mediated PPARγ degradation could provide valuable insights into adipose tissue biology and thermogenesis.

Biochemical analyses have established that m<sup>6</sup>A mRNA methylation is catalyzed by m<sup>6</sup>A methyltransferase complex (METTL3/14/WTAP) (Ping et al, 2014), recognized by its reader proteins (YTHDC1/2 and YTHDF1-3) (Frye et al, 2018; Yue et al, 2015), and removed by eraser proteins (FTO and ALKBH5) (Ding et al, 2025; Frye et al, 2018; Yue et al, 2015). However, the in vivo functions of these m<sup>6</sup>A related proteins and their molecular mechanisms vary across different cell types and tissues. One possible explanation for this variability is that the expression levels of m<sup>6</sup>A related proteins, as well as their target mRNAs and interacting proteins, differ between tissues and cell types. Another explanation is that m<sup>6</sup>A related proteins exert their function in both m<sup>6</sup>A-dependent and m<sup>6</sup>A-independent manners (Ding et al, 2025; Li et al, 2022; Li et al, 2021). Although both *Mettl3*-BKO and *Wtap*-BKO mice show similar phenotypes of larger and whitening iBAT (Wang et al, 2020; Wang et al, 2022b), *Ythdc1*-BKO mice display smaller and whitening iBAT. In the case of *Mettl3*-BKO and *Wtap*-BKO, these phenotypes are associated with reduced mRNA m<sup>6</sup>A modification and decreased expression of both *Pparg* and *Prdm16* transcripts (Wang et al, 2020; Wang et al, 2022b). The

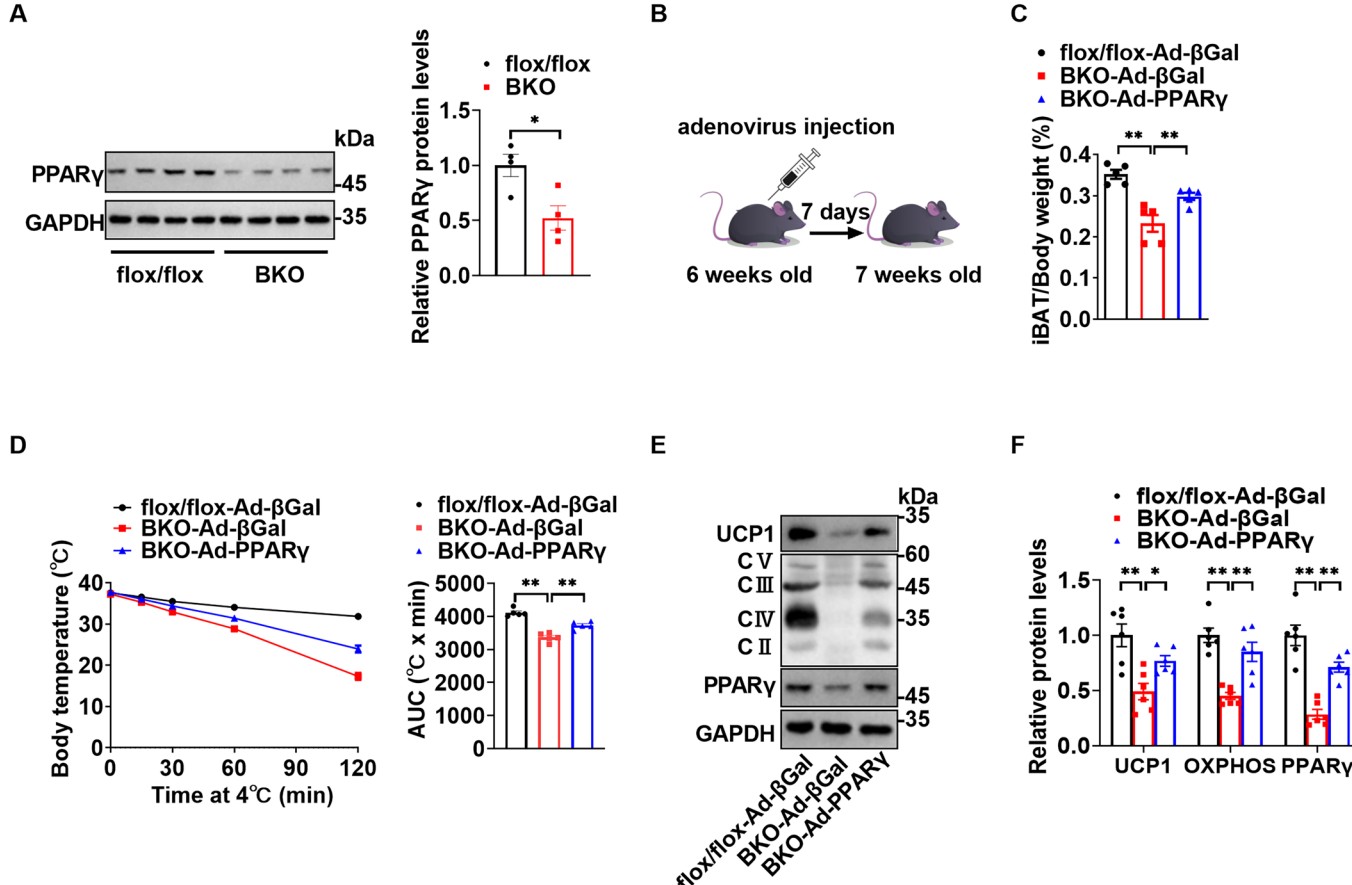

**Figure 5. Reduced PPARγ in iBATs of *Ythdc1*-BKO mice contributes to reduced iBAT mass and impaired thermogenesis.**

(A) PPARγ and GAPDH protein levels in iBATs of *Ythdc1*^flox/flox^ and *Ythdc1*-BKO mice at twenty days of age ($n = 4$ per group; $P = 0.0192$). (B) Schedule of rescue experiments. Re-expression of PPARγ in *Ythdc1*-BKO iBATs was achieved by multiple local injections of purified PPARγ adenovirus at 6 weeks of age. βGal adenovirus injection served as the control. We started the experiments one week later after injection. (C) Relative iBAT weights were measured ($n = 5$ per group; flox/flox-Ad-βGal VS BKO-Ad-βGal: $P < 0.0001$; BKO-Ad-βGal VS BKO-Ad-PPARγ: $P = 0.006758$). (D) Cold challenge experiments were performed ($n = 5$ per group; flox/flox-Ad-βGal VS BKO-Ad-βGal: $P < 0.0001$; BKO-Ad-βGal VS BKO-Ad-PPARγ: $P = 0.000583$). (E, F) UCP1, mitochondrial OXPHOS, and PPARγ protein levels were measured by immunoblotting and quantified using ImageJ ($n = 6$ per group; For UCP1, flox/flox-Ad-βGal VS BKO-Ad-βGal: $P = 0.000343$; BKO-Ad-βGal VS BKO-Ad-PPARγ: $P = 0.024087$; For OXPHOS, flox/flox-Ad-βGal VS BKO-Ad-βGal: $P < 0.0001$; BKO-Ad-βGal VS BKO-Ad-PPARγ: $P = 0.000583$; For PPARγ, flox/flox-Ad-βGal VS BKO-Ad-βGal: $P < 0.0001$; BKO-Ad-βGal VS BKO-Ad-PPARγ: $P = 0.000311$). Representative blot was shown in (E). *$P < 0.05$. **$P < 0.01$. Data represent the mean ± SEM. Differences between two groups were analyzed by unpaired two-tailed Student's *t* tests. Differences among more than two groups were analyzed by one-factor analysis of variance (ANOVA). *n* was the number of biologically independent mice. Source data are available online for this figure.

BAT-specific deletion of *Prdm16* results in age-dependent iBAT enlargement and whitening (Harms et al, 2014), indicating that reduced PRDM16 contributes to the observed phenotypes in *Mettl3*-BKO and *Wtap*-BKO mice. In contrast, both *Ythdc1*-FKO and *Ythdc1*-BKO significantly decreases PPARγ protein levels, while leaving *Pparg* mRNA levels unaffected, indicating that YTHDC1 regulates PPARγ independently of m6A recognition. *Ythdc1*-FKO mice show reduced WAT and iBAT, which is consistent with observations in *Pparg* knockout mice (He et al, 2003). The phenotype of smaller and whitening iBAT in both *Ythdc1*-FKO and *Ythdc1*-BKO mice is also consistent with the role of PPARγ in adipose tissue development, as *Pparg* knockout leads to smaller and whitening iBAT (He et al, 2003; Jones et al, 2005; Wang et al, 2013; Xiong et al, 2018). Importantly, restoring PPARγ expression in *Ythdc1*-BKO mice largely rescues these phenotypes,

suggesting that reduced PPARγ is the primary cause of the iBAT defects in *Ythdc1*-BKO mice. These findings emphasize the importance of studying m6A-related proteins within specific tissue contexts, as their roles can vary significantly depending on the tissue and molecular environment.

In conclusion, our study demonstrates that YTHDC1 plays a critical role in iBAT development and energy metabolism by interacting with PPARγ through its intrinsically disordered regions (IDRs). This interaction prevents ARIH2-mediated PPARγ degradation by blocking the association between ARIH2 and PPARγ. The loss or downregulation of YTHDC1 increases the exposure of PPARγ to ARIH2, leading to its degradation and resulting in impaired iBAT development and reduced iBAT size. This study reveals a novel molecular mechanism through which YTHDC1 regulates iBAT development and energy homeostasis.

**A**

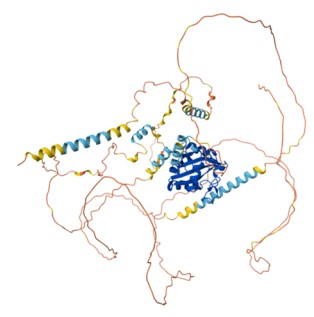

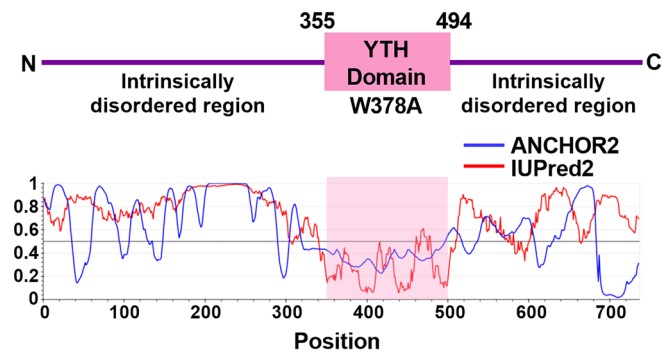

**B**

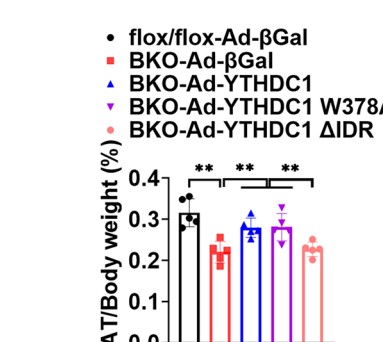

**C**

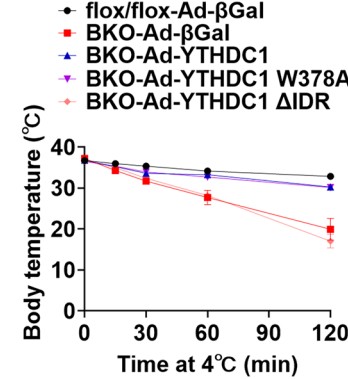

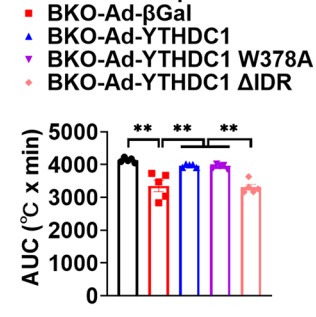

**D**

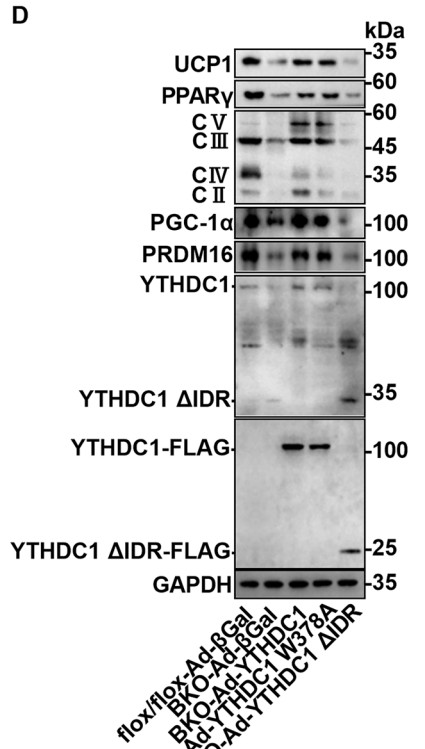

**E**

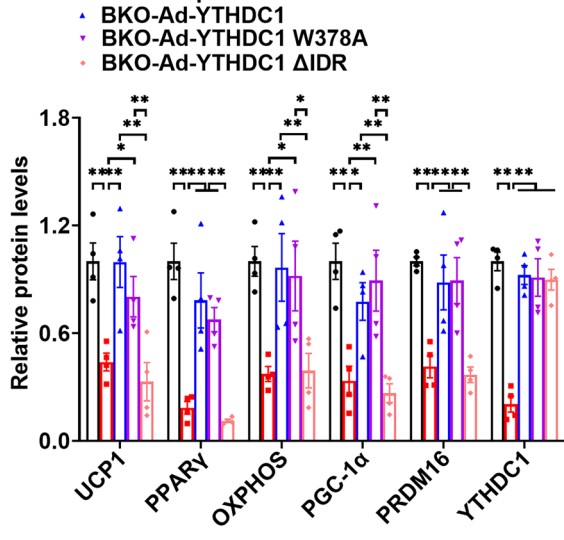

◄ **Figure 6. YTHDC1 regulates PPARγ expression, iBAT development, and thermogenesis through its intrinsic disorder region (IDR).**

(A) AlphaFold2, ANCHOR2, and IUPred2 predicted that YTHDC1 protein contains two larger intrinsic disorder regions and a YTH domain. (B–E) Re-expression of YTHDC1, YTHDC1 W378A, and YTHDC1 ΔIDR in *Ythdc1*-BKO iBATs was achieved by multiple local injections of purified respective adenovirus at 6 weeks of age. Ad-βGal adenovirus injection served as the control. We started the experiments one week later after injection. Relative iBAT weights were measured (B) ($n = 5$ per group; flox/flox-Ad-βGal VS BKO-Ad-βGal: $P < 0.0001$; BKO-Ad-βGal VS BKO-Ad-YTHDC1: $P = 0.0028$; BKO-Ad-βGal VS BKO-Ad-YTHDC1 W378A: $P = 0.0023$; BKO-Ad-YTHDC1 VS BKO-Ad-YTHDC1 ΔIDR: $P = 0.0069$; BKO-Ad-YTHDC1 W378A VS BKO-Ad-YTHDC1 ΔIDR: $P = 0.0059$). Thermogenesis was measured by cold exposure (C) ($n = 5$ per group; flox/flox-Ad-βGal VS BKO-Ad-βGal: $P < 0.0001$; BKO-Ad-βGal VS BKO-Ad-YTHDC1: $P = 0.0001$; BKO-Ad-βGal VS BKO-Ad-YTHDC1 W378A: $P = 0.0002$; BKO-Ad-YTHDC1 VS BKO-Ad-YTHDC1 ΔIDR: $P < 0.0001$; BKO-Ad-YTHDC1 W378A VS BKO-Ad-YTHDC1 ΔIDR: $P < 0.0001$). UCP1, PPARγ, PGC-1α, PRDM16, and mitochondrial oxidative phosphorylation (OXPHOS) levels were determined by immunoblotting and quantified using ImageJ (D, E) ($n = 5$ per group; For UCP1, flox/flox-Ad-βGal VS BKO-Ad-βGal: $P = 0.0021$; BKO-Ad-βGal VS BKO-Ad-YTHDC1: $P = 0.0022$; BKO-Ad-βGal VS BKO-Ad-YTHDC1 W378A: $P = 0.0293$; BKO-Ad-YTHDC1 VS BKO-Ad-YTHDC1 ΔIDR: $P = 0.0005$; BKO-Ad-YTHDC1 W378A VS BKO-Ad-YTHDC1 ΔIDR: $P = 0.0068$; For PPARγ, flox/flox-Ad-βGal VS BKO-Ad-βGal: $P < 0.0001$; BKO-Ad-βGal VS BKO-Ad-YTHDC1: $P = 0.00026$; BKO-Ad-βGal VS BKO-Ad-YTHDC1 W378A: $P = 0.0014$; BKO-Ad-YTHDC1 VS BKO-Ad-YTHDC1 ΔIDR: $P < 0.0001$; BKO-Ad-YTHDC1 W378A VS BKO-Ad-YTHDC1 ΔIDR: $P = 0.0004$; For OXPHOS, flox/flox-Ad-βGal VS BKO-Ad-βGal: $P = 0.0049$; BKO-Ad-βGal VS BKO-Ad-YTHDC1: $P = 0.0071$; BKO-Ad-βGal VS BKO-Ad-YTHDC1 W378A: $P = 0.0118$; BKO-Ad-YTHDC1 VS BKO-Ad-YTHDC1 ΔIDR: $P = 0.0087$; BKO-Ad-YTHDC1 W378A VS BKO-Ad-YTHDC1 ΔIDR: $P = 0.0144$; For PGC-1α, flox/flox-Ad-βGal VS BKO-Ad-βGal: $P = 0.0006$; BKO-Ad-βGal VS BKO-Ad-YTHDC1: $P = 0.0117$; BKO-Ad-βGal VS BKO-Ad-YTHDC1 W378A: $P = 0.0025$; BKO-Ad-YTHDC1 VS BKO-Ad-YTHDC1 ΔIDR: $P = 0.0047$; BKO-Ad-YTHDC1 W378A VS BKO-Ad-YTHDC1 ΔIDR: $P = 0.001$; For PRDM16, flox/flox-Ad-βGal VS BKO-Ad-βGal: $P = 0.0006$; BKO-Ad-βGal VS BKO-Ad-YTHDC1: $P = 0.0035$; BKO-Ad-βGal VS BKO-Ad-YTHDC1 W378A: $P = 0.003$; BKO-Ad-YTHDC1 VS BKO-Ad-YTHDC1 ΔIDR: $P = 0.0017$; BKO-Ad-YTHDC1 W378A VS BKO-Ad-YTHDC1 ΔIDR: $P = 0.0015$; For YTHDC1, flox/flox-Ad-βGal VS BKO-Ad-βGal: $P < 0.0001$; BKO-Ad-βGal VS BKO-Ad-YTHDC1: $P < 0.0001$; BKO-Ad-βGal VS BKO-Ad-YTHDC1 W378A: $P < 0.0001$; BKO-Ad-βGal VS BKO-Ad-YTHDC1 ΔIDR: $P < 0.0001$). Representative data was shown (D). *$P < 0.05$. **$P < 0.01$. Data represent the mean ± SEM. Differences among more than two groups were analyzed by one-factor analysis of variance (ANOVA). *n* was the number of biologically independent mice. Source data are available online for this figure.

# Methods

## Reagents and tools table

| Reagent/Resource | Reference or Source | Identifier or Catalog Number |
|---|---|---|
| **Experimental models** | | |
| *Ucp1*-iCre mice | Biocytogen | Stock #: 110134 |
| *Adipoq*-Cre mice | The Jackson Laboratory | Strain #:028020 |
| db/db mice | GemPharmatech | Strain #: T002407 |
| C57BL/6 mice | Charles river | |
| HEK293T cells (human) | ATCC | CRL-3216 |
| HEK293A cells (human) | Haixing Biosciences | TCH-C587 |
| DH5α | Tolobio | Cat#CC96102 |
| *Ythdc1*flox/flox mice | Cambridge-suda Genomic Resource Center | |
| **Recombinant DNA** | | |
| pDC316 | YouBio | Cat#VT1805 |
| pDC316-ZsGreen-shRNA | YouBio | Cat#VT2241 |
| **Antibodies** | | |
| Rabbit Anti-Mouse IgG | Cell Signaling Technology | Cat#58802 |
| Rabbit anti-YTHDC1 | Cell Signaling Technology | Cat#77422 |
| Rabbit anti-AKT | Cell Signaling Technology | Cat#9272 |
| Rabbit anti-p-AKT(T308) | Cell Signaling Technology | Cat#4056 |
| Rabbit anti-p-AKT (S473) | Cell Signaling Technology | Cat#9271 |
| Rabbit anti-UCP1 | Sigma | Cat#U6382 |
| Mouse anti-PGC-1a | Proteintech | Cat#66369-1-Ig |
| Mouse anti-β-Actin | Proteintech | Cat#60008-1-Ig |
| Rabbit anti-PPARγ | Proteintech | Cat#16643-1-AP |
| Mouse anti-GAPDH | Proteintech | Cat#60004-1-Ig |
| Mouse anti-Total OXPHOS | Abcam | Cat#ab110413 |

| Reagent/Resource | Reference or Source | Identifier or Catalog Number |
|---|---|---|
| Rabbit anti-PRDM16 | ABclonal | Cat#A11581 |
| Mouse anti-FLAG | Sigma | Cat#F1804 |
| Rabbit anti-HA | Proteintech | Cat#51064-2-AP |
| Rabbit anti-ARIH2 | Proteintech | Cat#15006-1-AP |
| Rabbit anti-ubiquitin | Proteintech | Cat#10201-2-AP |
| PerCP anti-mouse CD45.2 | BioLegend | Cat#109826 |
| PE anti-mouse F4/80 | BioLegend | Cat#123110 |
| PE anti-mouse NK-1.1 | BioLegend | Cat#108707 |
| APC anti-mouse CD3 | BioLegend | Cat#100236 |
| APC/Cyanine7 anti-mouse CD8b(ly-3) | BioLegend | Cat#126619 |
| APC anti-mouse/human CD11b | BioLegend | Cat#101212 |
| PE-Cyanine7 Anti-Mo CD4, eBioscience™ | Invitrogen | Cat#25-0041-82 |
| PE anti-mouse CD11c | BioLegend | Cat#117307 |
| PE/Cy7 anti-mouse/human CD45R/B220 | BioLegend | Cat#103222 |
| **Chemicals, Enzymes and other reagents** | | |
| Collagenase type II | Worthington | Cat#LS004177 |
| DispaseII | Roche | REF:04942078001 |
| FBS | TRINTY TEK | Cat#01010102 |
| DMEM high-glucose powder | Gibco | Cat#12100046 |
| penicillin/streptomycin | Viva Cell Biosciences | REF:C3420-0100 |
| Dexamethasone | Sigma-Aldrich | Cat#D1756 |
| Insulin | Sigma-Aldrich | Cat#I5500 |
| 3-Isobutyl-1-methylxanthine | Sigma-Aldrich | Cat#I5879 |
| Rosiglitazone | Sigma-Aldrich | Cat#R2408 |

| Reagent/Resource | Reference or Source | Identifier or Catalog Number |
|---|---|---|
| 3,3',5-Triiodo-L-thyronine | Sigma-Aldrich | Cat#T2877 |
| Indomethacin | Sigma-Aldrich | Cat#I7378 |
| BSA | Solarbio | Cat#1A8020 |
| F12 medium | Cytiva | Cat#SH30023.01 |
| Cycloheximide | Sigma-Aldrich | Cat#66-81-9 |
| Super sensitive ELC luminescent solution | Clinx | Cat#1810202 |
| Tripure Isolation Reagent | Roche | Cat#94015120 |
| MG132 | Biotechne | Cat#1748 |
| Leupeptin | sigma-Aldrich | Cat#L2884 |
| SYBR Green Mix | Roche | Cat#4913914001 |
| RIPA buffer | Solarbio | Cat#R0020 |
| CL316,243 | Sigma-Aldrich | Cat#C5976 |
| TriPure Isolation Reagent | Roche | Cat#94015120 |
| Random Primers | Promega | Cat#C1181 |
| M-MLV reverse transcriptase | Promega | Cat#M1701 |
| Trans2K DNA Marker | Transgen Biotech | Cat#BM101 |
| Blue Plus II Protein Marker | Transgen Biotech | Cat#DM111 |
| Blue Plus IV Protein Marker | Transgen Biotech | Cat#DM131 |
| Oil Red O | Sigma-Aldrich | Cat#O0625 |
| **Software** | | |
| Graphpad Prism 8 | Graphpadsoftwave | Graphpad.com |
| ImageJ | National Institutes of Health | |
| **Other** | | |
| High-Fat Diet | Changzhou SYSE Bio-Tec | Cat#PD6001 |
| Promethion | Sable Systems | |
| Glucometer | Ren et al, 2017 | |
| Insulin ELISA kits | EZassay | Cat#MS100 |
| Rectal probe | Phyritemp | MODEL NO. 7001HT |
| Roche LightCycler® 480 II | Roche | 5015278001 |
| BD LSRFortessa™ Cell Analyzer | BD Biosciences | Cat#647794L6 |
| NovaSeq 6000 platform | Illumina | |
| LSM 880 | Zeiss | |

## Animal experiments

All animal experiments were conducted in strict compliance with the Guide for the Care and Use of Laboratory Animals and were approved by the Institutional Animal Care and Use Committee or Animal Experimental Ethics Committee of Harbin Institute of Technology (HIT/IACUC). Mice were maintained in a controlled environment with a 12-h light/dark cycle, at a temperature of $24 \pm 2\,°C$, and $50\% \pm 10\%$ humidity, and were provided with a

standard chow diet and free access to water. For diet-induced obesity studies, mice were fed a high-fat diet (HFD) containing 60% fat (PD6001, Changzhou SYSE Bio-Tec. Co., Ltd.).

The $Ythdc1^{flox/flox}$ mice were provided by the Cambridge-suda Genomic Resource Center. The exons 5–7 of the $Ythdc1$ gene were flanked by two loxp sites. $Ucp1$-iCre mice, which have an IRES-Cre inserted between exon 6 and the 3'UTR allowing simultaneous $Ucp1$ and $iCre$ expression at lower levels, were generated as previously described (Li et al, 2017). BAT-specific $Ythdc1$ knockout mice ($Ythdc1$-BKO) were produced by crossing $Ythdc1^{flox/flox}$ mice with $Ucp1$-iCre mice. Fat-specific $Ythdc1$ knockout mice ($Ythdc1$-FKO) were produced by crossing $Ythdc1^{flox/flox}$ mice with $Adipoq$-Cre mice (Strain #: 028020, Jackson laboratory). To minimize the effects of subjective bias, mice of each genotype were mixed and housed randomly.

For rescue experiments, 6-week-old male $Ythdc1$-BKO mice received local iBAT injections of equal amounts of purified adenoviruses (Ad-YTHDC1, Ad-YTHDC1 W378A, Ad-YTHDC1 ΔIDR, Ad-PPARγ, Ad-shArih2-1, and Ad-shArih2-2). Male $Ythdc1^{flox/flox}$ mice were injected with purified Ad-βGal or Ad-shScramble adenoviruses in equivalent amounts. Experiments commenced one week after adenoviral injection. The sequences targeted by shRNA were as follows: shArih2-1: GCGCTACCTCTT-TAGGGACTAT; shArih2-2: GCTGGATGTGTCTAGGAGATT. The targeted sequence for shScramble was described previously (Wang et al, 2020). Reagents and tools were used in this study were shown in Reagents and Tools Table.

## Energy metabolism measurement

For metabolic studies, male mice were individually housed in metabolic cages (Promethion, Sable Systems, Las Vegas, NV) with unrestricted access to food and water. Oxygen consumption and $CO_2$ production were monitored over a 72-h period, alongside simultaneous measurement of physical activity and food intake. Data were collected and analyzed using MetaScreen-Data Collection Software (V2.3.15) and Expedata-P Data Analysis Software (V1.9.17).

## Glucose tolerance tests and insulin tolerance tests

For glucose tolerance tests, male mice were fasted for 6 h and then injected intraperitoneally with D-glucose (1 g/kg body weight). For insulin tolerance tests, mice were similarly fasted for 6 h and injected intraperitoneally with human insulin (1 U/kg body weight; Lily). Blood glucose levels were measured from the tail vein at specified time points using a glucometer (Ren et al, 2017). Blood samples were collected from the orbital sinus, and serum insulin levels were quantified using insulin ELISA kits (MS100, EZassay).

## In vivo insulin stimulation assay

For the in vivo insulin stimulation assay, 25-week-old HFD-fed male $Ythdc1^{flox/flox}$ and $Ythdc1$-BKO mice were fasted for 20–24 h. After fasting, the mice were anesthetized and administered insulin (1 unit/kg body weight) via the inferior vena cava. Livers were then isolated, homogenized in lysis buffer (R0020, Solarbio), and subjected to immunoblotting with antibodies against

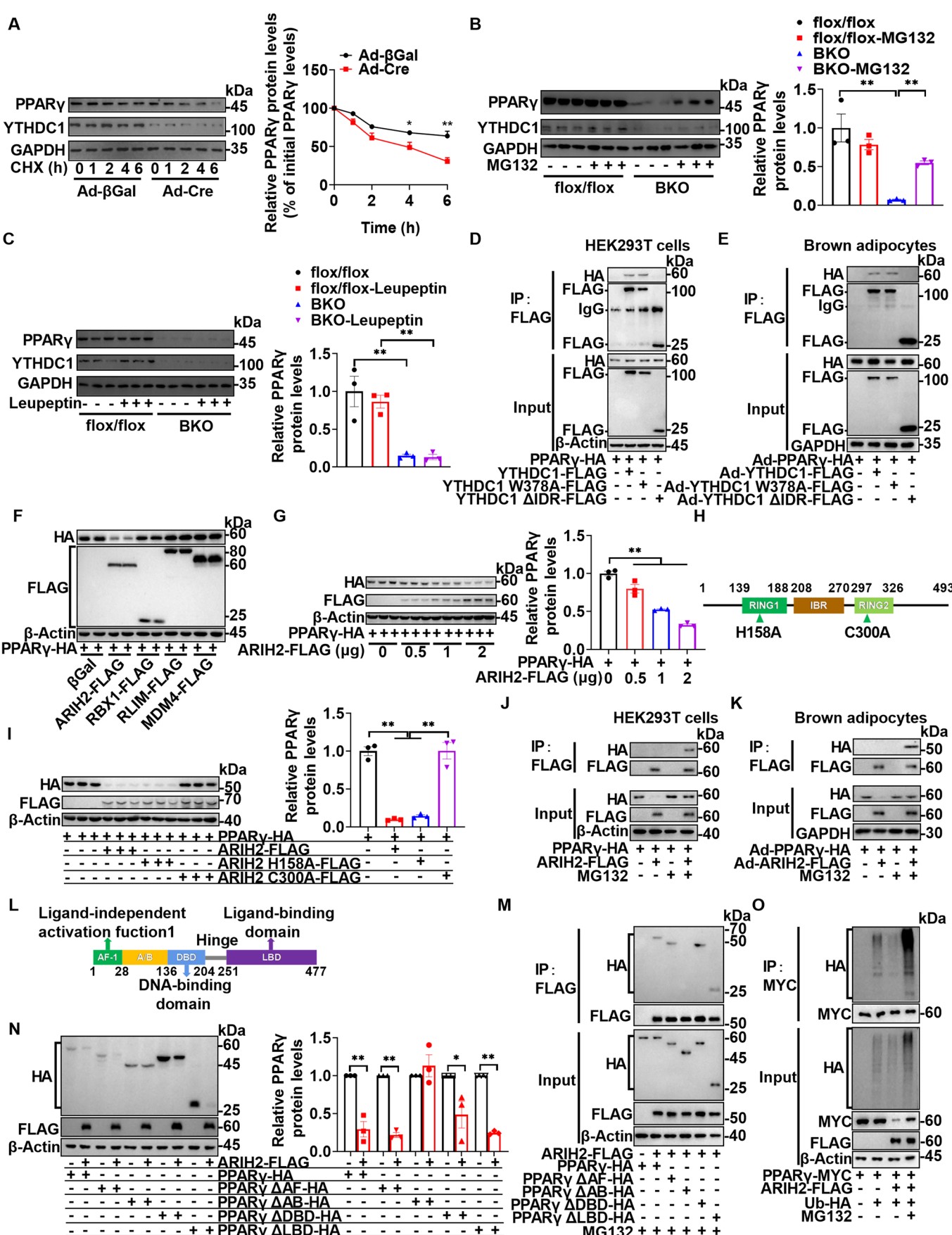

◄ **Figure 7.   YTHDC1 regulates protein stability of PPARγ through its IDR with the involvement of E3 ligase ARIH2.**

(A) Brown fat stromal-vascular fraction (SVF) was isolated from 6-week-old *Ythdc1*flox/flox mice and cultured in growth medium. 100% confluent cells were induced in induction medium for three days. Cells were then maintained in differentiation medium for two days. Differentiated brown adipocytes were infected with Cre- or βGal-expressing adenovirus for 48 h. The infected cells were then were treated with cycloheximide (CHX, 2 µg/ml) for indicated periods. Cells were then harvested for immunoblotting analysis of PPARγ, YTHDC1, and GAPDH protein levels. The relative PPARγ protein levels were represented as the percentage of the band densities at 0 h. Quantifications of relative PPARγ protein levels were from five independent experiments ($n = 5$ per group; 1h: $P = 0.0540$; 2 h: $P = 0.0551$; 4 h: $P = 0.0247$; 6 h: $P = 0.0026$). (B, C) The iBATs were dissected from 6-week-old *Ythdc1*flox/flox and *Ythdc1*-BKO mice and cut into 10 mg pieces. Pieces of iBATs were randomly divided into two groups. One group were treated with or without MG132 (100 µM) (B) and the other group were treated with or without leupeptin (100 µM) (C) at 37 °C for 6 h. PPARγ, YTHDC1, and GAPDH protein levels were measured by immunoblotting. Quantifications of relative PPARγ protein levels were from three independent mice ($n = 3$ per group; In (B), flox/flox VS BKO: $P = 0.0001$; BKO VS BKO-MG132: $P = 0.0083$; In (C), flox/flox VS BKO: $P = 0.0006$; flox/flox-Leupeptin VS BKO-Leupeptin: $P = 0.0016$). (D) PPARγ-HA expression vector was co-transfected with or without YTHDC1-FLAG, YTHDC1 W378A-FLAG, or YTHDC1ΔIDR-FLAG expression vector in HEK293T cells for 30 h. Cell lysates were extracted from these cells in RIPA buffer. These lysates were immunoprecipitated with anti-FLAG beads and then immunoblotted with anti-HA, anti-FLAG, or anti-β-Actin antibodies. The samples were derived from the same experiment and the blots were processed in parallel. (E) Primary brown adipocytes were differentiated from brown fat stromal-vascular fraction (SVF). Ad-PPARγ-HA adenovirus was co-infected with Ad-βGal, Ad-YTHDC1-FLAG, Ad-YTHDC1W378A-FLAG, or Ad-YTHDC1ΔIDR-FLAG adenovirus for 48 h. Cell lysates were extracted from these cells in RIPA buffer. These lysates were immunoprecipitated with anti-FLAG beads and then immunoblotted with anti-HA, anti-FLAG, or anti-GAPDH antibodies. The samples were derived from the same experiment and the blots were processed in parallel. (F) PPARγ-HA expression vector was co-transfected with βGal, ARIH2-FLAG, RBX1-FLAG, RLIM-FLAG, or MDM4-FLAG expression vector in HEK293T cells for 30 h. Cell lysates were extracted from these cells in RIPA buffer. These lysates were immunoblotted with anti-HA, anti-FLAG, or anti-β-Actin antibodies. The samples were derived from the same experiment and the blots were processed in parallel. (G) PPARγ-HA expression vector was co-transfected with different doses of ARIH2-FLAG (0, 0.5, 1, and 2 µg) expression vector in HEK293T cells for 30 h. Cell lysates were extracted from these cells in RIPA buffer. These lysates were immunoblotted with anti-HA, anti-FLAG, or anti-β-Actin antibodies. The samples were derived from the same experiment and the blots were processed in parallel ($n = 3$ per group; 0 VS 0.5: $P = 0.0025$; 0 VS 1: $P < 0.0001$; 0 VS 2: $P < 0.0001$). (H) RING1, IBR, and RING2 domains of ARIH2 were shown. (I) PPARγ-HA expression vector was co-transfected with or without ARIH2-FLAG, ARIH2 H158A-FLAG, or ARIH2 C300A-FLAG expression vector in HEK293T cells for 30 h. Cell lysates were extracted from these cells in RIPA buffer. These lysates were immunoblotted with anti-HA, anti-FLAG, or anti-β-Actin antibodies. The samples were derived from the same experiment and the blots were processed in parallel ($n = 3$ per group; PPARγ-HA VS PPARγ-HA + ARIH2-FLAG: $P < 0.0001$; PPARγ-HA VS PPARγ-HA + ARIH2 H158A-FLAG: $P < 0.0001$; PPARγ-HA + ARIH2-FLAG VS PPARγ-HA + ARIH2 C300A-FLAG: $P < 0.0001$; PPARγ-HA + ARIH2 H158A-FLAG VS PPARγ-HA + ARIH2 C300A-FLAG: $P < 0.0001$). (J) PPARγ-HA expression vector was co-transfected with or without ARIH2-FLAG expression vector in HEK293T cells for 30 h. Cells were treated with or without MG132 (20 µM) for 6 h. Cell lysates were extracted from these cells in RIPA buffer. These lysates were immunoprecipitated with anti-FLAG beads and then immunoblotted with anti-HA, anti-FLAG, or anti-β-Actin antibodies. The samples were derived from the same experiment and the blots were processed in parallel. (K) Primary brown adipocytes were differentiated from brown fat stromal-vascular fraction (SVF). Ad-PPARγ-HA adenovirus was co-infected with Ad-βGal or Ad-ARIH2-FLAG adenovirus overnight. Cells were treated with or without MG132 (20 µM) for 6 h. Cell lysates were extracted from these cells in RIPA buffer. These lysates were immunoprecipitated with anti-FLAG beads and then immunoblotted with anti-HA, anti-FLAG, or anti-GAPDH antibodies. The samples were derived from the same experiment and the blots were processed in parallel. (L) Domains in PPARγ were shown. (M) Different forms of PPARγ-HA (PPARγ, PPARγΔAF, PPARγΔAB, PPARγΔDBD, and PPARγΔLBD) were co-transfected with or without ARIH2-FLAG expression vector in HEK293T cells for 30 h. Cells were treated with MG132 (20 µM) for 6 h. Cell lysates were extracted from these cells in RIPA buffer. These lysates were immunoprecipitated with anti-FLAG beads and then immunoblotted with anti-HA, anti-FLAG, or anti-β-Actin antibodies. The samples were derived from the same experiment and the blots were processed in parallel. (N) Different forms of PPARγ-HA (PPARγ, PPARγΔAF, PPARγΔAB, PPARγΔDBD, and PPARγΔLBD) were co-transfected with or without ARIH2-FLAG expression vector in HEK293T cells for 30 h. Cell lysates were extracted from these cells in RIPA buffer. These lysates were immunoblotted with anti-HA, anti-FLAG, or anti-β-Actin antibodies. The samples were derived from the same experiment and the blots were processed in parallel ($n = 3$ per group; PPARγ-HA VS PPARγ-HA + ARIH2-FLAG: $P = 0.0022$; PPARγΔAF-HA VS PPARγΔAF + ARIH2-FLAG: $P < 0.0001$; PPARγΔAB-HA VS PPARγΔAB + ARIH2-FLAG: $P = 0.4176$; PPARγΔDBD VS PPARγΔDBD + ARIH2-FLAG: $P = 0.0447$; PPARγΔLBD-HA VS PPARγΔLBD + ARIH2-FLAG: $P < 0.0001$). (O) PPARγ-MYC was co-transfected with or without FLAG-ARIH2 or HA-ubiquitin expression vector in HEK293T cells for 30 h. Cells were treated with or without MG132 (20 µM) for 6 h. Cell lysates were extracted from these cells in RIPA buffer. These lysates were immunoprecipitated with anti-MYC beads and then immunoblotted with anti-HA, anti-MYC, or anti-β-Actin antibodies. The samples were derived from the same experiment and the blots were processed in parallel. All the cell culture experiments were repeated three times with similar results. *$P < 0.05$. **$P < 0.01$. Data represent the mean ± SEM. Differences between two groups were analyzed by unpaired two-tailed Student's $t$ tests. Differences among more than two groups were analyzed by one-factor analysis of variance (ANOVA). $n$ was the number of biologically independent cell samples. Source data are available online for this figure.

phosphorylated AKT (pSer473 and pThr308) and total AKT (Cell Signaling Technology).

## Cold-stress experiments

For cold exposure experiments, individual male mice were housed in separate cages in a cold room (4 °C) with free access to water. Another group of male mice was maintained at room temperature (22 °C), serving as control. Core body temperature was measured at specified time points using a rectal probe (7001HT, Phyritemp) as shown previously (Wang et al, 2020; Wang et al, 2022b).

## Chronic CL-316,243 treatment

Male *Ythdc1*flox/flox and *Ythdc1*-BKO mice were injected intraperitoneally with CL-316,243 at 1 mg/kg body weight or equal volume of saline daily for 4 days. On day 5, the mice were sacrificed without additional injection.

## Quantitative real-time PCR (qPCR)

Total RNA was extracted using TriPure Isolation Reagent (94015120, Roche). First-strand cDNA synthesis was performed using Random Primers and M-MLV reverse transcriptase (M1701, Promega) (Li et al, 2018; Ren et al, 2017). Gene expression was measured using SYBR Green Mix (4913914001, Roche) and analyzed using a Roche LightCycler 480 real-time PCR system (Roche, Mannheim, Germany). Gene expression was normalized to the house-keeping gene 36B4. Primer sequences were shown in Appendix Table S1.

## Culture of primary brown and white adipocytes and adenovirus infection

Primary brown adipocyte culture was followed a published protocol (Wang et al, 2020). The interscapular brown fat pad was dissected from 6-week-old male *Ythdc1*flox/flox or C57BL/6 wild-type mice,

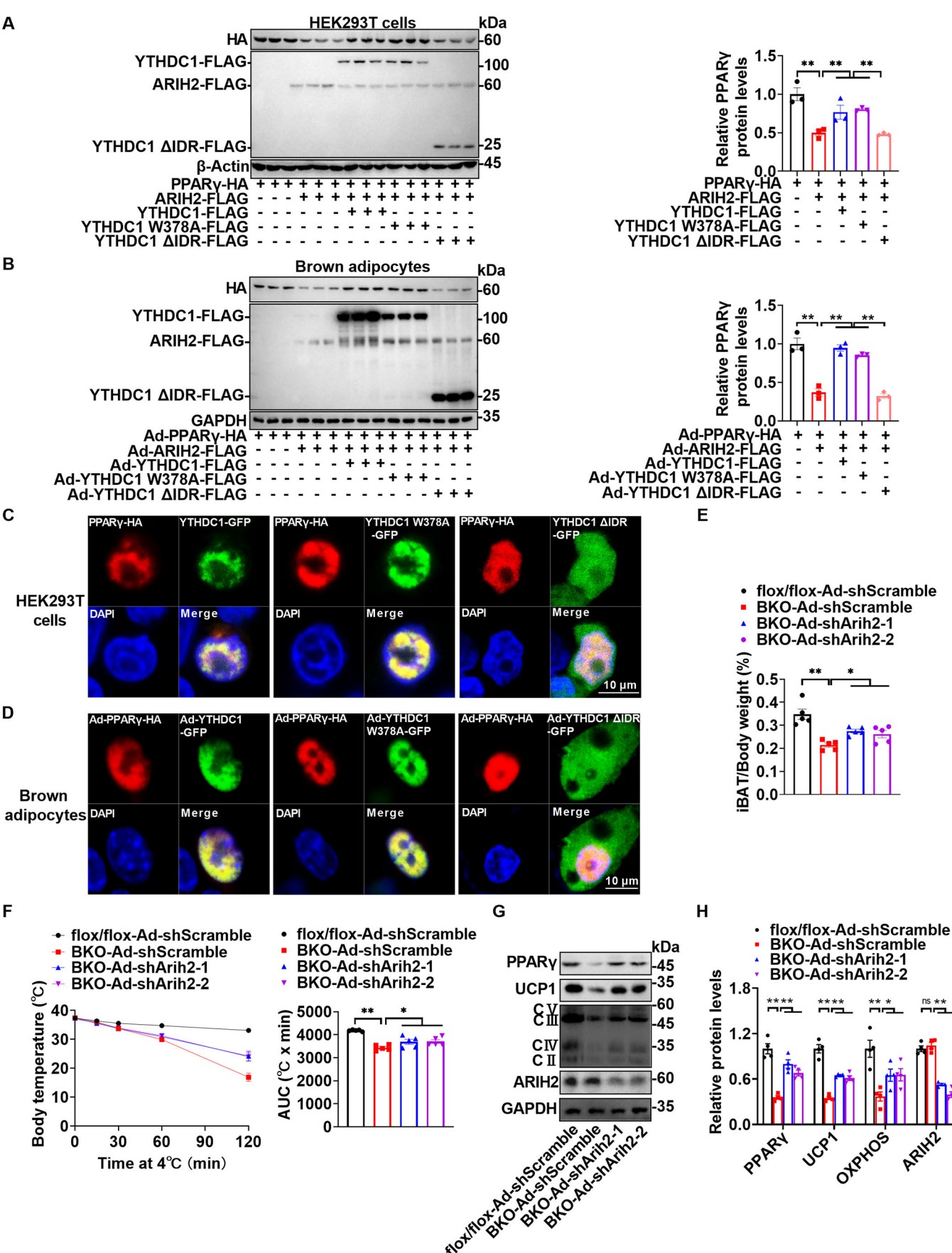

◄ **Figure 8. ARIH2 is required for degradation of PPARγ in *Ythdc1*-BKO mice.**

(A) PPARγ-HA expression vector was co-transfected with YTHDC1-FLAG, YTHDC1 W378A-FLAG, or YTHDC1ΔIDR-FLAG expression vectors in HEK293T cells for 30 h. These cells were then transfected with or without ARIH2-FLAG expression vector. Thirty hours later, cell lysates were extracted from these cells in RIPA buffer. These lysates were immunoblotted with anti-HA, anti-FLAG, or anti-β-Actin antibodies. The samples were derived from the same experiment and the blots were processed in parallel ($n = 3$ per group; PPARγ-HA VS PPARγ-HA + ARIH2-FLAG: $P = 0.0001$; PPARγ-HA + ARIH2-FLAG VS PPARγ-HA + ARIH2-FLAG + YTHDC1-FLAG: $P = 0.0085$; PPARγ-HA + ARIH2-FLAG VS PPARγ-HA + ARIH2-FLAG + YTHDC1 W378A-FLAG: $P = 0.0041$; PPARγ-HA + ARIH2-FLAG + YTHDC1-FLAG VS PPARγ-HA + ARIH2-FLAG + YTHDC1ΔIDR-FLAG: $P = 0.0058$; PPARγ-HA + ARIH2-FLAG + YTHDC1 W378A-FLAG VS PPARγ-HA + ARIH2-FLAG + YTHDC1ΔIDR-FLAG: $P = 0.0028$). (B) Primary brown adipocytes were differentiated from brown fat stromal-vascular fraction (SVF). Ad-PPARγ-HA adenovirus was co-infected with Ad-βGal, Ad-YTHDC1-FLAG, Ad-YTHDC1W378A-FLAG, or Ad-YTHDC1ΔIDR-FLAG adenovirus for 24 h. Cells were then infected with or without Ad-ARIH2-FLAG adenovirus. Twenty-four hours later, cell lysates were extracted from these cells in RIPA buffer. These lysates were immunoblotted with anti-HA, anti-FLAG, or anti-GAPDH antibodies. The samples were derived from the same experiment and the blots were processed in parallel ($n = 3$ per group; Ad-PPARγ-HA VS Ad-PPARγ-HA + Ad-ARIH2-FLAG: $P < 0.0001$; Ad-PPARγ-HA + Ad-ARIH2-FLAG VS Ad-PPARγ-HA + Ad-ARIH2-FLAG + Ad-YTHDC1-FLAG: $P < 0.0001$; Ad-PPARγ-HA + Ad-ARIH2-FLAG VS Ad-PPARγ-HA + Ad-ARIH2-FLAG + Ad-YTHDC1 W378A-FLAG: $P < 0.0001$; Ad-PPARγ-HA + Ad-ARIH2-FLAG + Ad-YTHDC1-FLAG VS Ad-PPARγ-HA + Ad-ARIH2-FLAG + Ad-YTHDC1ΔIDR-FLAG: $P < 0.0001$; Ad-PPARγ-HA + Ad-ARIH2-FLAG + Ad-YTHDC1 W378A-FLAG VS Ad-PPARγ-HA + Ad-ARIH2-FLAG + Ad-YTHDC1ΔIDR-FLAG: $P < 0.0001$). (C) PPARγ-HA expression vector was co-transfected with YTHDC1-GFP, YTHDC1 W378A-GFP, or YTHDC1ΔIDR-GFP expression vectors in HEK293T cells for 30 h. Cells were immunostained with anti-HA antibody. The co-localization of PPARγ-HA and different forms of YTHDC1-GFP was analyzed using confocal microscopy. (D) Primary brown adipocytes were differentiated from brown fat stromal-vascular fraction (SVF). Ad-PPARγ-HA adenovirus was co-infected with Ad-βGal, Ad-YTHDC1-GFP, Ad-YTHDC1W378A-GFP, or Ad-YTHDC1ΔIDR-GFP adenovirus for 48 h. Cells were immunostained with anti-HA antibody. The co-localization of HA-PPARγ and different forms of YTHDC1-GFP was analyzed using confocal microscopy. (E–G) iBAT-specific knockdown of *Arih2* in *Ythdc1*-BKO iBATs was achieved by multiple local injections of purified Ad-shArih2-1 or Ad-shArih2-2 adenovirus at 6 weeks of age. Ad-shScramble adenovirus injection served as the control. We started the experiments one week later after injection. Relative iBAT weights were measured (E) ($n = 5$ per group; flox/flox-Ad-shScramble VS BKO-Ad-shScramble: $P < 0.0001$; BKO-Ad-shScramble VS BKO-Ad-shArih2-1: $P = 0.0113$; BKO-Ad-shScramble VS BKO-Ad-shArih2-2: $P = 0.0389$). Thermogenesis was measured by cold exposure (F) ($n = 5$ per group; flox/flox-Ad-shScramble VS BKO-Ad-shScramble: $P < 0.0001$; BKO-Ad-shScramble VS BKO-Ad-shArih2-1: $P = 0.0239$; BKO-Ad-shScramble VS BKO-Ad-shArih2-2: $P = 0.0181$). PPARγ, UCP1, mitochondrial oxidative phosphorylation (OXPHOS), ARIH2, and GAPDH levels were determined by immunoblotting and quantified using ImageJ (G, H) ($n = 4$ per group; For PPARγ, flox/flox-Ad-shScramble VS BKO-Ad-shScramble: $P < 0.0001$; BKO-Ad-shScramble VS BKO-Ad-shArih2-1: $P < 0.0001$; BKO-Ad-shScramble VS BKO-Ad-shArih2-2: $P = 0.0006$; For UCP1, flox/flox-Ad-shScramble VS BKO-Ad-shScramble: $P < 0.0001$; BKO-Ad-shScramble VS BKO-Ad-shArih2-1: $P < 0.0001$; BKO-Ad-shScramble VS BKO-Ad-shArih2-2: $P = 0.0001$; For OXPHOS, flox/flox-Ad-shScramble VS BKO-Ad-shScramble: $P = 0.0002$; BKO-Ad-shScramble VS BKO-Ad-shArih2-1: $P = 0.0392$; BKO-Ad-shScramble VS BKO-Ad-shArih2-2: $P = 0.0358$; For ARIH2, BKO-Ad-shScramble VS BKO-Ad-shArih2-1: $P < 0.0001$; BKO-Ad-shScramble VS BKO-Ad-shArih2-2: $P < 0.0001$). Representative data (G) and quantified data (H) were shown. All the cell culture experiments were repeated three times with similar results. n was the number of biologically independent cell samples. Data represent the mean ± SEM. Differences among more than two groups were analyzed by one-factor analysis of variance (ANOVA). *$P < 0.05$. **$P < 0.01$. Source data are available online for this figure.

minced, and then digested for 20–30 min at 37 °C in PBS containing 10 mM $CaCl_2$, 1.5 mg/ml Collagenase type II, and 1.4 U/ml DispaseII. The digested tissue was filtered through a 40-μm cell strainer to remove large debris and centrifuged for 10 min at $1000 \times g$ to pellet the stromal-vascular fraction (SVF) cells. SVF cells were resuspended in complete culture medium (DMEM with 10% FBS and penicillin/streptomycin) and plated onto collagen-coated 24-well plates. For preadipocyte differentiation, cells were grown to confluence (Day 0) and then induced with DMEM containing 2 μg/mL dexamethasone, 1 μM insulin, 0.5 mM isobutylmethylxanthine, 1 μM rosiglitazone, 1 nM $T_3$, 62.5 μM indomethacin, and 10% FBS. After 3 days (Day 3), cells were maintained in DMEM containing 1 μM insulin, 1 nM $T_3$, and 10% FBS until harvesting, generally on days 6–7 post-differentiation. All cell culture chemicals were purchased from Sigma-Aldrich.

Methods for primary white adipocyte culture are shown below. Male mice were euthanized by cervical dislocation, and inguinal white adipose tissue (iWAT) and epididymal white adipose tissue (eWAT) were collected and minced. The minced tissue was added to 7–10 mL of DMEM containing 1 mg/mL collagenase and 1% BSA and digested at 37 °C for 25 min for iWAT and 40 min for eWAT. The mixture was gently triturated, and digestion was terminated by adding 0.5–1 mL of serum. The digested tissue was filtered through a 70-μm cell strainer, centrifuged at $500 \times g$ for 5 min, and the supernatant was discarded. The pellet was resuspended in F12 medium and incubated at 37 °C for 4 h, after which the medium was replaced with F12 containing 10 ng/mL FGF and cultured until the cells reached confluence. For cell induction, the cells were treated with F12 medium containing 0.5 μM IBMX, 1 μM Dex, and 10 μg/mL insulin for 3–4 days.

Primary white adipocyte differentiation was achieved by culturing the cells in F12 medium containing 10 μg/mL insulin for 4–6 days.

For adenoviral infection, differentiated brown or white adipocytes were infected with equal amounts of YTHDC1, YTHDC1 W378A, YTHDC1 ΔIDR, PPARγ, ARIH2, Cre, or βGal-expressing adenoviruses for 48 h. Cells were then harvested for immunoprecipitation or immunoblotting analysis. For protein stability assay, infected cells were then treated with or without cycloheximide (CHX, 2 μg/mL) for the indicated time points, followed by harvesting for immunoblotting analysis.

## Fluorescence-activated cell sorting (FACS) analysis

The stromal-vascular fraction (SVF) cells were isolated from iBATs of male *Ythdc1*^flox/flox^ and *Ythdc1*-FKO mice. SVF cells were stained with indicated antibodies as shown below. One group of SVF cells were stained with PerCP anti-mouse CD45.2 (109826, BioLegend) and PE anti-mouse F4/80 (123110, BioLegend). One group of SVF cells were stained with PerCP anti-mouse CD45.2 (109826, BioLegend), PE anti-mouse NK-1.1 (108707, BioLegend), APC anti-mouse CD3 (100236, BioLegend), APC/Cyanine7 anti-mouse CD8b(ly-3) (126619, BioLegend), and PE-Cyanine7 Anti-Mo CD4, eBioscience™ (25-0041-82, Invitrogen). Another group of SVF cells were stained with PerCP anti-mouse CD45.2 (109826, BioLegend), APC anti-mouse/human CD11b (101212, BioLegend), PE anti-mouse CD11c (117307, BioLegend), and PE/Cy7 anti-mouse/human CD45R/B220 (103222, BioLegend). The dates were collected with a BD FACSAria III flow cytometer with three laser lines (405, 488, and 633 nm).

## Transfection, immunoprecipitation, and immunoblotting

HEK293T cells (CRL-11268, ATCC) were co-transfected with PPARγ-HA expression vector, along with or without YTHDC1-FLAG, YTHDC1 W378A-FLAG, or YTHDC1ΔIDR-FLAG expression vectors, for 30 h. HEK293T cells were authenticated and tested for mycoplasma contamination. Additionally, primary brown adipocytes were differentiated from the stromal-vascular fraction (SVF) of brown adipose tissue. These cells were co-infected with Ad-PPARγ-HA adenovirus and either Ad-βGal, Ad-YTHDC1-FLAG, Ad-YTHDC1 W378A-FLAG, or Ad-YTHDC1ΔIDR-FLAG adenovirus for 48 h. Cell lysates were prepared using RIPA buffer (R0020, Solarbio), and immunoprecipitation was performed using anti-FLAG beads. The immunoprecipitates were subsequently immunoblotted with anti-HA or anti-FLAG antibodies.

For immunoblotting, cells or tissues were homogenized in L-RIPA lysis buffer. Proteins were separated by SDS-PAGE and transferred to membranes, followed by immunoblotting with the specified antibodies. Detection was carried out using enhanced chemiluminescence (ECL). The antibody information and dilutions used were as follows: YTHDC1 (77422, Cell Signaling Technology), 1:3000; AKT (9272, Cell Signaling Technology), 1:5000; p-AKT(T308) (4056, Cell Signaling Technology), 1:5000; p-AKT (S473) (9271, Cell Signaling Technology), 1:5000; UCP1 (U6382, Sigma), 1:5000; PGC-1a (66369-1-lg, Proteintech), 1:1000; β-Actin (60008-1-lg, Proteintech), 1:5000; PPARγ (16643-1-AP, Proteintech), 1:3000; GAPDH (60004-1-lg, Proteintech), 1:5000; Total OXPHOS (ab110413, Abcam), 1:2000; PRDM16 (A11581, ABclonal), 1:1000; FLAG (F1804, Sigma), 1:5000; HA (51064-2-AP, Proteintech), 1:5000; ARIH2 (15006-1-AP, Proteintech), 1:3000; ubiquitin (10201-2-AP, Proteintech), 1:4000.

## RNA-seq

Total RNA was extracted using Tripure Isolation Reagent (94015120, Roche) from iBATs of *Ythdc1*$^{flox/flox}$ and *Ythdc1*-BKO mice at 8 weeks old ($n = 3$ per group). RNA-seq was performed on the Illumina NovaSeq 6000 platform. Paired-end clean reads were aligned to the mouse reference genome (Ensemble_GRCm38/mm10) using TopHat (version 2.0.12). The aligned reads were quantified for mRNA expression using HTSeq-count (version 0.6.1). The RNA-seq analysis were performed by staffs at Novogene who were blinded to the experimental groups. Blinding was not relevant to the other experiments in mice or cells because mice or cells had to be genotyped by PCR.

## Statistical analysis

All data are presented as means ± S.E. The Shapiro-Wilk test was employed to assess the normality of the data. When all groups were normally distributed ($p > 0.05$), the parametric two-tailed Student's t tests were used to detect the statistical differences between the two groups. The parametric one-factor analysis of variance (ANOVA), and Tukey was used to detect the statistical differences among four groups. When at least one of the two groups were not normally distributed ($p < 0.05$), the non-parametric Mann–Whitney test was adopted to compare the statistical differences between the two groups. The non-parametric Kruskal–Wallis and Dunn's was adopted to compare the statistical differences among four groups. All statistical analyses were performed using GraphPad Prism 8.0 (GraphPad Software Inc., San Diego, CA, USA). ANOVA and LSD-t test analyses were performed using SPSS 21.0 (SPSS Inc, Chicago, IL). Statistical significance was set at $P < 0.05$. Significance levels are indicated as follows: *$P < 0.05$. **$P < 0.01$.

## Data availability

Data supporting the findings of this study are included in the article and Supplementary Information. All additional data are available from the corresponding authors upon reasonable request. RNA-seq data have been deposited into Gene Expression Omnibusdatabase (www.ncbi.nlm.nih.gov/geo) (GEO) database under accession number GSE273849.

The source data of this paper are collected in the following database record: biostudies:S-SCDT-10_1038-S44318-025-00460-x.

## Peer review information

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

## Acknowledgements

This study was supported by the National Natural Science Foundation of China Grant (32371198 and 82401019), Fundamental Research Funds for the Central Universities (HIT.OCEF.2025001), the Heilongjiang Chunyan Team (CYCX24020) and GDAS' Project of Science and Technology Development and Young Talent Project of GDAS (2024GDASZH-2024010102 and 2024GDASQNRC-0104). We thank Novogene for assistance in RNA-seq experiment. We thank Ming Gao and Xin Wang for the help of the immunoblotting and confocal microscopy experiments.

## Author contributions

**Lihua Wang**: Formal analysis; Investigation; Visualization. **Yuqin Wang**: Formal analysis; Investigation; Visualization. **Kaixin Ding**: Formal analysis; Investigation; Visualization. **Zhenzhi Li**: Formal analysis; Investigation; Visualization. **Zhipeng Zhang**: Formal analysis; Investigation; Visualization. **Xinzhi Li**: Investigation. **Yue Song**: Investigation. **Liwei Xie**: Resources; Methodology. **Zheng Chen**: Conceptualization; Supervision; Writing—original draft; Writing—review and editing.

Source data underlying figure panels in this paper may have individual authorship assigned. Where available, figure panel/source data authorship is listed in the following database record: biostudies:S-SCDT-10_1038-S44318-025-00460-x.

## Disclosure and competing interests statement

The authors declare no competing interests.

