## [Peer Review File · The EMBO Journal]

YTHDC1 promotes postnatal brown adipose tissue development and thermogenesis by stabilizing PPAR γ

Lihua Wang, Yuqin Wang, Kaixin Ding, Zhenzhi Li, Zhipeng Zhang, Xinzhi Li, Yue Song, Liwei Xie, and Zheng Chen

Corresponding author: Zheng Chen (chenzheng@hit.edu.cn)

Review Timeline:

Submission Date:	3rd Nov 24
Editorial Decision:	17th Dec 24
Revision Received:	14th Mar 25
Editorial Decision:	9th Apr 25
Revision Received:	17th Apr 25
Accepted:	22nd Apr 25

Editor: Daniel Klimmeck

Transaction Report:

Dear Dr Chen,

Thank you again for the submission of your manuscript (EMBOJ-2024-119515) to The EMBO Journal. As mentioned earlier, your study was assessed by three reviewers with expertise in central control of body metabolism and adipocyte biology, whose comments are enclosed below.

As you will see from the experts' reports, the referees acknowledge the analysis and potential interest and value of your findings. However, they also express important issues regarding the completeness of your study and physiological relevance of the results, which need to be addressed thoroughly to make them supportive of publication in the EMBO Journal. Further, the reviewers raise a number of issues related to the presentation of the findings, additional controls and improved methods annotation required, statistics applied and overall discussion of related literature, that would need to be conclusively addressed to achieve the level of robustness and clarity needed for The EMBO Journal.

Given the overall interest stated and broader angle of your findings, we are able to invite you to revise your manuscript experimentally to address the referees' comments. I need to stress though that we do require strong support from the referees on a revised version of the study in order to move on to publication of the work.

I would appreciate if you could contact me during the next weeks for exchange e.g. a video call to discuss your perspective on the comments and potential plan for revisions.

Please feel free to contact me if you have any questions or need further input on the referee comments.

When submitting your revised manuscript, please carefully review the instructions below.

Please feel free to approach me any time should you have additional questions related to this.

Thank you for the opportunity to consider your work for publication.

I look forward to your revision.

Kind regards,

Daniel Klimmeck

Daniel Klimmeck, PhD
Senior Editor
The EMBO Journal

Instruction for the preparation of your revised manuscript:

- 1) a .docx formatted version of the manuscript text (including legends for main figures, EV figures and tables). Please make sure that the changes are highlighted to be clearly visible.
- 2) individual production quality figure files as .eps, .tif, .jpg (one file per figure).
- 3) a .docx formatted letter INCLUDING the reviewers' reports and your detailed point-by-point response to their comments. As part of the EMBO Press transparent editorial process, the point-by-point response is part of the Review Process File (RPF), which will be published alongside your paper.
- 4) a complete author checklist, which you can download from our author guidelines ([https://wol-prod-cdn.literatumonline.com/pb-assets/embo-site/Author Checklist%20-%20EMBO%20J-1561436015657.xlsx](https://wol-prod-cdn.literatumonline.com/pb-assets/embo-site/Author%20Checklist%20-%20EMBO%20J-1561436015657.xlsx)). Please insert information in the checklist that is also reflected in the manuscript. The completed author checklist will also be part of the RPF.

6) It is mandatory to include a 'Data Availability' section after the Materials and Methods. Before submitting your revision, primary datasets produced in this study need to be deposited in an appropriate public database, and the accession numbers and database listed under 'Data Availability'. Please remember to provide a reviewer password if the datasets are not yet public (see <https://www.embopress.org/page/journal/14602075/authorguide#datadeposition>).

7) Our journal encourages inclusion of *data citations in the reference list* to directly cite datasets that were re-used and obtained from public databases. Data citations in the article text are distinct from normal bibliographical citations and should directly link to the database records from which the data can be accessed. In the main text, data citations are formatted as follows: "Data ref: Smith et al, 2001" or "Data ref: NCBI Sequence Read Archive PRJNA342805, 2017". In the Reference list, data citations must be labeled with "[DATASET]". A data reference must provide the database name, accession number/identifiers and a resolvable link to the landing page from which the data can be accessed at the end of the reference. Further instructions are available at .

8) At EMBO Press we ask authors to provide source data for the main and EV figures. Our source data coordinator will contact you to discuss which figure panels we would need source data for and will also provide you with helpful tips on how to upload and organize the files.

Numerical data can be provided as individual .xls or .csv files (including a tab describing the data). For 'blots' or microscopy, uncropped images should be submitted (using a zip archive or a single pdf per main figure if multiple images need to be supplied for one panel). Additional information on source data and instruction on how to label the files are available at .

9) We replaced Supplementary Information with Expanded View (EV) Figures and Tables that are collapsible/expandable online (see examples in <https://www.embopress.org/doi/10.15252/embj.201695874>). A maximum of 5 EV Figures can be typeset. EV Figures should be cited as 'Figure EV1, Figure EV2" etc. in the text and their respective legends should be included in the main text after the legends of regular figures.

11) For data quantification: please specify the name of the statistical test used to generate error bars and P values, the number (n) of independent experiments (specify technical or biological replicates) underlying each data point and the test used to calculate p-values in each figure legend. The figure legends should contain a basic description of n, P and the test applied. Graphs must include a description of the bars and the error bars (s.d., s.e.m.).

We realize that it is difficult to revise to a specific deadline. In the interest of protecting the conceptual advance provided by the work, we recommend a revision within 3 months (17th Mar 2025). Please discuss the revision progress ahead of this time with the editor if you require more time to complete the revisions.

Referee #1:

In the current manuscript, the authors generated conditional *Ythdc1* knockout mice by using UCP1 promoter. The authors report that these mice had defects in BAT development upon cold exposure and these mice were cold intolerant with decreased BAT mass with lower expression of PPAR, UCP1 and other BAT-selective genes. The authors report that YTHDC1, a known m6A reader, interacts with PPAR to prevent its interaction with an E3 ubiquitin ligase, ARIH2, for protection from proteasomal degradation.

The YTHDC1 function in PPAR stability or its YTHDC1 interaction with an E3 ligase ARIH2 have not been shown previously, making the current study new and worth reporting. However, as listed below, there are some major and minor points to be addressed.

1. The authors did not examine or demonstrate ubiquitination of PPAR. The ubiquitin antibody should be used in the immunoblotting to demonstrate ubiquitination of PPAR and changes.
2. The authors employed primary brown adipocytes in addition to HEK293 cells to demonstrate specific interactions. However, even in primary brown adipocytes, both PPAR and its putative interacting proteins were overexpressed for Co-IP. The authors need to demonstrate interaction of endogenous proteins.
3. The authors indicate changes in BAT selective genes by RNA-seq. The authors also describe changed in expression of other genes associated with immune response activation, defense response to pathogens, phagocytosis etc. It is possible that, in addition to brown adipocytes, the composition of SVF that contain macrophages and other immune cell types may have changed. This aspect was not addressed in this study.
4. The authors did not perform any morphological examinations of BAT, only indicating changes in BAT weight and gene expression. In Figures 1 and 2, thorough examination of brown adipose tissue sections for lipid staining, immunostaining need to be performed.
5. It is unclear why the total body weight did not change when BKO mice were maintained at room temperature (not thermoneutrality), when these mice undergoing thermogenesis. Only ectopic fat in liver was shown in Figure 2. At minimum, white fat mass should be documented in these BKO mice.
6. Control mice on high fat diet for 16 weeks did not appear to be significantly insulin resistant, this is puzzling. Moreover, while the authors claim these BKO mice to be insulin resistant, GTT and ITT did not appear to be significantly affected by *Ythdc1*-BKO, although insulin level is higher. A minor point: ITT should be plotted by glucose concentrations not by % change.
7. The authors need to show overexpression by adenoviral YTHDC1. Localization of UATHDC1 IDR was not clear. GFP signal appears to be diffused in cells in Figure 7. It is possible IDR was not localized to nucleus preventing interaction with PPAR. (However, the authors used total lysates for Co-IP for interaction experiments).
8. A minor point: There is duplicate references of Wang Y, Gao M, Zhu F, Li X, Yang Y, Yan Q, Jia L, Xie L, Chen Z (2020a)

Referee #2:

Chen and associates demonstrate that the m6A reader protein YTHDC1 is a key regulator of postnatal BAT development. They show that YTHDC1 interacts with PPAR γ via its intrinsically disordered region (IDR) to protect PPAR γ from degradation. This interaction prevents ARIH2, an E3 ubiquitin ligase, from binding, ubiquitinating and degrading PPAR γ . Absence of YTHDC1 in iBAT via UCP promoter directed Cre leads to a significant reduction in PPAR γ and PRDM16 as well as mitochondrial protein expression. As shown by others, absence of PPAR γ in iBAT has an overall systemic effect on energy metabolism.

Comments: The experiments are well executed with very convincing data.

Be interesting to know what protects PPAR gamma from degradation in white adipose tissue. Also, are the white genes affected by knock out of YTHDC1 in WAT following CL treatment or do they require lower levels of PPAR gamma to maintain their expression compared to BAT genes?

Referee #3:

This study focused on the function of YTHDC1 in brown adipocytes. They found that YTHDC1 interacts with PPAR γ and protects it from degradation. This finding is novel and contributes largely to the fundamentals of adipocyte biology. Overall, the experiments are well-planned, and the data are solid and supportive of the molecular mechanisms of how YTHDC1 regulates PPAR γ protein level.

I have a few comments on the correlation of the molecular function of YTHDC1 in brown adipose tissue.

1. The author stated that YTHDC1 is the key regulator of postnatal BAT development. However, according to the literature, BAT is developed embryonically, and it is unclear how many brown adipocytes are generated postnatally. Cold exposure also doesn't lead to significant brown adipogenesis.
2. Besides adipogenesis, PPAR γ is a key transcription factor maintaining adipocyte function; knocking out PPAR γ causes adipocyte death. Instead of inhibiting brown adipogenesis, the cold intolerance phenotype after YTHDC1 deletion is more likely to be due to impairments in mature brown adipocyte function and even cell death. It would be nice to test these possibilities.
3. PPAR γ is also a master regulator of white adipogenesis and mature adipocyte function; it would be nice to perform some experiments on white adipocytes to confirm if the mechanism is brown-specific or in all types of adipocytes.
4. It would be nice to show if there is any phenotype on beige adipocytes after cold exposure.

Minor: The figure legend is too long. Some content could be moved to the method section.

Response to Reviewers' comments

Referee #1:

In the current manuscript, the authors generated conditional *Ythdc1* knockout mice by using UCP1 promoter. The authors report that these mice had defects in BAT development upon cold exposure and these mice were cold intolerant with decreased BAT mass with lower expression of PPAR, UCP1 and other BAT-selective genes. The authors report that YTHDC1, a known m6 A reader, interacts with PPAR to prevent its interaction with an E3 ubiquitin ligase, ARIH2, for protection from proteasomal degradation.

The YTHDC1 function in PPAR stability or its YTHDC1 interaction with an E3 ligase ARIH2 have not been shown previously, making the current study new and worth reporting. However, as listed below, there are some major and minor points to be addressed.

1. The authors did not examine or demonstrate ubiquitination of PPAR. The ubiquitin antibody should be used in the immunoblotting to demonstrate ubiquitination of PPAR and changes.

Response: Thank you for your valuable comment. We appreciate your suggestion to examine the ubiquitination of PPAR γ . We agree that this would provide important insights into its regulation. We have conducted additional experiments to assess the ubiquitination status of PPAR γ using an anti-ubiquitin antibody, and have included the findings in the revised manuscript. As shown in Fig. S7C, ARIH2 promotes ubiquitination of PPAR γ .

C

We hope these new data address your concerns and improve the clarity of our manuscript. Thank you once again for your valuable feedback.

2. The authors employed primary brown adipocytes in addition to HEK293 cells to demonstrate specific interactions. However, even in primary brown adipocytes, both PPAR and its putative interacting proteins were overexpressed for Co-IP. The authors need to demonstrate interaction of endogenous proteins.

Response: Thank you for your valuable comment. To address your comments, we have conducted additional Co-IP experiments to measure the interaction of endogenous proteins in iBAT. As shown in Fig. S6A, PPAR γ interacts with YTHDC1 in iBAT.

A

We hope these new data address your concerns and improve the clarity of our manuscript. Thank you once again for your valuable feedback.

3. The authors indicate changes in BAT selective genes by RNA-seq. The authors also describe changed in expression of other genes associated with immune response activation, defense response to pathogens, phagocytosis etc. It is possible that, in addition to brown adipocytes, the composition of SVF that contain macrophages and other immune cell types may have changed. This aspect was not addressed in this study.

Response: Thank you for your valuable comment. To address your comments, we have conducted additional experiments. RNA-seq and RT-qPCR showed that Both *Ythdc1*-BKO and *Ythdc1*-FKO increased the expression of inflammation-related genes in iBAT (Fig. 4H-I). We also performed fluorescence activated cell sorting (FACS) analysis on SVF to measure the changes of immune cells in FKO and flox/flox SVFs. As shown in Fig. 4J-L and S4A-C, total immune (CD45.2⁺) cells, macrophage (CD45.2⁺ F4/80⁺), total T (CD45.2⁺ CD3⁺) cells, CD4 T (CD45.2⁺ CD3⁺ CD4⁺) cells, NK (CD45.2⁺ NK1.1⁺ CD3⁻) cells, B (CD45.2⁺ B220⁺ CD11c⁻) cells were significantly increased in SVF from iBAT of *Ythdc1*-FKO mice, while dendritic cells

(CD45.2⁺CD11c⁺CD11b⁻) and CD8 T (CD45.2⁺ CD3⁺ CD8⁺) cells were not changed in iBAT of *Ythdc1*-FKO mice. These data demonstrate that the deletion of *Ythdc1* causes immune cell infiltration in iBAT, leading to higher expression of inflammation-related genes.

A

B

C

We hope these new data address your concerns and improve the clarity of our manuscript. Thank you once again for your valuable feedback.

4. The authors did not perform any morphological examinations of BAT, only indicating changes in BAT weight and gene expression. In Figures 1 and 2, thorough examination of brown adipose tissue sections for lipid staining, immunostaining need to be performed.

Response: Thank you for your valuable comment. To address your comments, we have conducted morphological examinations of iBAT including oil red O staining and immunostaining of UCP1. As shown in Fig. S3A-B, S3D, S3G-H, and S3J, H&E and Oil Red O staining showed higher lipid accumulation in *Ythdc1*-BKO iBAT, and immunostaining of UCP1 showed lower UCP1 expression in *Ythdc1*-BKO iBAT.

We hope these new data address your concerns and improve the clarity of our manuscript. Thank you once again for your valuable feedback.

5. It is unclear why the total body weight did not change when BKO mice were maintained at room temperature (not thermoneutrality), when these mice undergoing thermogenesis. Only ectopic fat in liver was shown in Figure 2. At minimum, white fat mass should be documented in these BKO mice.

Response: Thank you for your valuable comment. Food intake is similar between flox/flox and BKO mice (Fig. S3G). Reduced energy expenditure may promote HFD-induced obesity in BKO mice. However, *Ythdc1*-BKO mice gained similar body weight to *Ythdc1*^{flox/flox} mice during HFD feeding (Fig.2B), indicating that *Ythdc1*-BKO mice are not more susceptible to HFD-induced obesity. One possible reason is the increased physical activity in *Ythdc1*-BKO mice (Fig. S3F). How *Ythdc1*-BKO increases physical activity is unknown. We discussed this issue in the discussion section.

To further address your comments, we have provided weights of tissues including white adipose tissues (iWAT and eWAT) in HFD-fed BKO and flox/flox mice. As shown in Fig. S3L, the weights of iWAT and eWAT were similar between *Ythdc1*^{flox/flox} and *Ythdc1*-BKO mice.

We hope these new data address your concerns and improve the clarity of our manuscript. Thank you once again for your valuable feedback.

6. Control mice on high fat diet for 16 weeks did not appear to be significantly insulin resistant, this is puzzling. Moreover, while the authors claim these BKO mice to be insulin resistant, GTT and ITT did not appear to be significantly affected by *Ythdc1*-BKO, although insulin level is higher. A minor point: ITT should be plotted by glucose concentrations not by % change.

Response: Thank you for your thoughtful comments and for raising these important points. We acknowledge that high-fat diet (HFD) typically induces insulin resistance. Therefore, we generally use a low dose of glucose (1g/kg) for the glucose tolerance test (GTT) and a higher dose of insulin (1U/kg) for the insulin tolerance test (ITT) for HFD-fed mice. In contrast, for normal chow (NC)-fed mice, we use a higher dose of glucose (2g/kg) for the GTT and a lower dose of insulin (0.75U/kg) for the ITT. It

appears that the control mice on the high-fat diet for 16 weeks did not exhibit significant insulin resistance, which may be attributed to the experimental conditions, including the lower glucose dose used for the GTT and the higher insulin dose used for the ITT.

As shown in Fig. 2H-L, *Ythdc1*-BKO mice exhibited slightly, but statistically significant, increased glucose intolerance and insulin resistance compared to flox/flox mice. We have clarified this in the revised manuscript to provide a clearer explanation of these results.

We appreciate your suggestion regarding the representation of ITT data. We have re-plotted the ITT results using absolute glucose concentrations rather than percentage changes, as recommended. The revised figure has been included in the manuscript.

Thank you again for your constructive feedback, which will help strengthen our manuscript.

7. The authors need to show overexpression by adenoviral YTHDC1. Localization of UAthDC1 IDR was not clear. GFP signal appears to be diffused in cells in Figure 7. It is possible IDR was not localized to nucleus preventing interaction with PPAR. (However, the authors used total lysates for Co-IP for interaction experiments).

Response: Thank you for your valuable comment. To address your comments, we have conducted additional experiments by overexpression using Ad-FLAG-YTHDC1 or Ad-FLAG-YTHDC1 Δ IDR and performed immunostaining. Ad-FLAG-YTHDC1 and Ad-FLAG-YTHDC1 Δ IDR shows similar localization with Ad-GFP-YTHDC1 and Ad-GFP-YTHDC1 Δ IDR. YTHDC1 formed nuclear puncta with PPAR γ in an IDR-dependent manner in HEK293T cells (Fig. 8C and S8B), primary brown adipocytes (Fig. 8D and S8C), and primary white adipocytes (Fig. S8D-E), suggesting

that YTHDC1, through its IDR, physically interacts with PPAR γ to shield it from ARIH2 binding and subsequent degradation. Both Ad-FLAG-YTHDC1 Δ IDR and Ad-GFP-YTHDC1 Δ IDR show both nuclear and cytosolic localization in HEK293T cells (Fig. 8C and S8B), primary brown adipocytes (Fig. 8D and S8C), and primary white adipocytes (Fig. S8D-E).

These is a slight co-localization of FLAG-YTHDC1 Δ IDR and HA-PPAR γ in nuclei. However, co-IP experiments showed no interaction between FLAG-YTHDC1 Δ IDR and HA-PPAR γ (Fig. 7D-E and S6B).

We hope these new data address your concerns and improve the clarity of our manuscript. Thank you once again for your valuable feedback.

8. A minor point: There is duplicate references of

Wang Y, Gao M, Zhu F, Li X, Yang Y, Yan Q, Jia L, Xie L, Chen Z (2020a)

Response: Thank you for your valuable feedback. We appreciate your attention to detail and your suggestion. We have revised the references.

Referee #2:

Chen and associates demonstrate that the m6A reader protein YTHDC1 is a key regulator of postnatal BAT development. They show that YTHDC1 interacts with PPAR γ via its intrinsically disordered region (IDR) to protect PPAR γ from degradation. This interaction prevents ARIH2, an E3 ubiquitin ligase, from binding, ubiquitinating and degrading PPAR γ . Absence of YTHDC1 in iBAT via UCP promoter directed Cre leads to a significant reduction in PPAR γ and PRDM16 as well as mitochondrial protein expression. As shown by others, absence of PPAR γ in iBAT has an overall systemic effect on energy metabolism.

Comments: The experiments are well executed with very convincing data.

Be interesting to know what protects PPAR gamma from degradation in white adipose tissue. Also, are the white genes affected by knock out of YTHDC1 in WAT following CL treatment or do they require lower levels of PPAR gamma to maintain their expression compared to BAT genes?

Response: Thank you for your kind words and thoughtful comments. We appreciate your interest in further exploring the underlying mechanisms regulating PPAR gamma stability in white adipose tissue (WAT) and the potential effects of YTHDC1 knockout (KO) on gene expression of white adipose tissue. During the preparation of this BAT paper, we generated fat, including both BAT and WAT, -specific *Ythdc1* knockout mice by using adipoq promoter driven Cre transgenic mice to investigate the role of YTHDC1 in white fat tissue. Our initial plan is to present the data in a separate paper. We have now combined these data into one paper. As shown in Fig. 4B-E, adipose-specific *Ythdc1* knockout mice display smaller iBAT, eWAT, and iWAT, which is likely due to the decreased PPAR γ protein levels, but not the mRNA levels

(Fig. S5B-D). Some PPAR γ targeted genes such as *Fabp4*, *Plin2*, *Cd36*, *Glut4*, and *Adipoq* were significantly downregulated in both iWAT and eWAT of *Ythdc1*-FKO mice (Fig. 4F), which further leads to smaller WAT (Fig. 4B). *Ythdc1*-FKO mice also showed impaired thermogenesis after cold exposure (Fig. 4G), which was likely attributed to downregulated UCP1 levels (Fig. 4C). Both *Ythdc1*-BKO and *Ythdc1*-FKO increased the expression of inflammation-related genes in iBAT (Fig. 4H-I). To further test whether deletion of *Ythdc1* in iBAT causes immune cell infiltration, we performed fluorescence activated cell sorting (FACS) analysis on SVF isolated from the iBAT of *Ythdc1*^{flox/flox} and *Ythdc1*-FKO mice. As shown in Fig. 4J-L and Fig. S4A-C, total immune (CD45.2⁺) cells, macrophage (CD45.2⁺ F4/80⁺), total T (CD45.2⁺ CD3⁺) cells, CD4 T (CD45.2⁺ CD3⁺ CD4⁺) cells, NK (CD45.2⁺ NK1.1⁺ CD3⁻) cells, B (CD45.2⁺ B220⁺ CD11c⁻) cells were significantly increased in SVF from iBAT of *Ythdc1*-FKO mice, while dendritic cells (CD45.2⁺ CD11c⁺ CD11b⁻) and CD8 T (CD45.2⁺ CD3⁺ CD8⁺) cells were not changed in iBAT of *Ythdc1*-FKO mice. These data demonstrate that the deletion of *Ythdc1* causes immune cell infiltration in iBAT, leading to higher expression of inflammation-related genes. These results suggest that YTHDC1 is important for the development of both BAT and WAT through the involvement of PPAR γ .

These data also indicate that deletion of *Ythdc1* in white adipose tissue also decreases the protein stability of PPAR γ . E3 ubiquitin ligase ARIH2 is also likely involved in this process (Fig. S6B, S8A, and S8D-E).

Thank you once again for your constructive feedback.

Referee #3:

This study focused on the function of YTHDC1 in brown adipocytes. They found that YTHDC1 interacts with PPAR γ and protects it from degradation. This finding is novel and contributes largely to the fundamentals of adipocyte biology. Overall, the experiments are well-planned, and the data are solid and supportive of the molecular mechanisms of how YTHDC1 regulates PPAR γ protein level.

I have a few comments on the correlation of the molecular function of YTHDC1 in brown adipose tissue.

1. The author stated that YTHDC1 is the key regulator of postnatal BAT development. However, according to the literature, BAT is developed embryonically, and it is unclear how many brown adipocytes are generated postnatally. Cold exposure also doesn't lead to significant brown adipogenesis.

Response: Thank you for your thoughtful comment. We appreciate your attention to the timing of brown adipose tissue (BAT) development and your insightful suggestion for clarification regarding the role of YTHDC1 in this process.

While it is well-established that BAT primarily develops during embryonic stages, several studies over the past few decades have demonstrated that BAT undergoes significant postnatal development, particularly in response to environmental cues such as temperature. For instance, the weight of inguinal BAT (iBAT) increases approximately 5-10 times from neonatal to adult stages, highlighting the postnatal development of BAT. Specifically, environmental temperature plays a critical role in this process, with cold exposure being particularly important for the maturation and functional activation of brown adipocytes. In neonatal rodents, exposure to relatively low temperatures (22-24°C, compared to the uterine temperature of 37°C)

significantly promotes *Ucp1* expression and enhances BAT growth (PMID: 2497735). Conversely, a thermoneutral environment (36°C) impairs postnatal BAT development (PMID: 2497735, 8660292). Moreover, exposure to cold temperatures (16°C) accelerates BAT growth and increases mitochondrial content in neonatal rats, further supporting the importance of temperature in postnatal BAT development (PMID: 1966247).

Thank you again for your valuable feedback, which help us refine the manuscript.

2. Besides adipogenesis, PPAR γ is a key transcription factor maintaining adipocyte function; knocking out PPAR γ causes adipocyte death. Instead of inhibiting brown adipogenesis, the cold intolerance phenotype after YTHDC1 deletion is more likely to be due to impairments in mature brown adipocyte function and even cell death. It would be nice to test these possibilities.

Response: Thank you for your insightful comments. We appreciate your suggestion to explore the possibility that the cold intolerance phenotype observed after YTHDC1 deletion may be due to impairments in mature brown adipocyte function or even cell death, rather than a direct effect on brown adipogenesis.

We agree that PPAR γ is crucial not only for adipogenesis but also for maintaining the function and survival of mature adipocytes. To test the hypothesis that YTHDC1 deletion impairs mature brown adipocyte function or leads to cell death, we have conducted additional experiments examining the viability of brown adipocytes in *Ythdc1*-BKO mice. Specifically, we assess markers of cell death, such as TUNEL staining and levels of cleaved caspase-3. We did not observe any positive cells or cleaved caspase-3 in either *Ythdc1*^{flox/flox} or *Ythdc1*-BKO iBAT, indicating that *Ythdc1*-BKO does not induce severe cell death in iBAT.

These results help clarify whether the cold intolerance phenotype is primarily due to defects in postnatal development of iBAT and brown adipocyte function, and we have incorporated the results into the revised manuscript that “We did not detect any TUNEL-positive cells or cleaved caspase 3 in both *Ythdc1*^{flox/flox} and *Ythdc1*-BKO iBAT, indicating that *Ythdc1*-BKO does not induce cell apoptosis in iBAT.”

Thank you again for your valuable suggestion, which helps strengthen our understanding of the role of YTHDC1 in brown adipocyte function.

3. PPAR γ is also a master regulator of white adipogenesis and mature adipocyte function; it would be nice to perform some experiments on white adipocytes to confirm if the mechanism is brown-specific or in all types of adipocytes.

Response: Thank you for your thoughtful comments. We appreciate your interest in further exploring the underlying functions and mechanisms of YTHDC1 in white adipose tissue (WAT). During the preparation of this BAT paper, we generated fat, including both BAT and WAT, -specific *Ythdc1* knockout mice by using adipog promoter driven Cre transgenic mice to investigate the role of YTHDC1 in white fat tissue. Our initial plan is to present the data in a separate paper. We have now combined these data into one paper. As shown in Fig. 4B-E, adipose-specific *Ythdc1* knockout mice display smaller iBAT, eWAT, and iWAT, which is likely due to the decreased PPAR γ protein levels, but not the mRNA levels (Fig. S5B-D). Some PPAR γ targeted genes such as *Fabp4*, *Plin2*, *Cd36*, *Glut4*, and *Adipoq* were significantly downregulated in both iWAT and eWAT of *Ythdc1*-FKO mice (Fig. 4F), which further leads to smaller WAT (Fig. 4B). *Ythdc1*-FKO mice also showed impaired thermogenesis after cold exposure (Fig. 4G), which was likely attributed to downregulated UCP1 levels (Fig. 4C). Both *Ythdc1*-BKO and *Ythdc1*-FKO increased the expression of inflammation-related genes in iBAT (Fig. 4H-I). To further test whether deletion of *Ythdc1* in iBAT causes immune cell infiltration, we performed fluorescence activated cell sorting (FACS) analysis on SVF isolated from the iBAT of *Ythdc1*^{flox/flox} and *Ythdc1*-FKO mice. As shown in Fig. 4J-L and Fig. S4A-C, total immune (CD45.2⁺) cells, macrophage (CD45.2⁺ F4/80⁺), total T (CD45.2⁺ CD3⁺) cells, CD4 T (CD45.2⁺ CD3⁺ CD4⁺) cells, NK (CD45.2⁺ NK1.1⁺ CD3⁻) cells, B (CD45.2⁺ B220⁺ CD11c⁻) cells were significantly increased in SVF from iBAT of *Ythdc1*-FKO mice, while dendritic cells (CD45.2⁺ CD11c⁺ CD11b⁻) and CD8 T (CD45.2⁺ CD3⁺ CD8⁺) cells were not changed in iBAT of *Ythdc1*-FKO mice. These data demonstrate that the deletion of *Ythdc1* causes immune cell infiltration in iBAT, leading to higher expression of inflammation-related genes. These results suggest that YTHDC1 is important for the development of both BAT and WAT through the involvement of PPAR γ .

These data also indicate that deletion of *Ythdc1* in white adipose tissue also decreases the protein stability of PPAR γ . E3 ubiquitin ligase ARIH2 is also likely involved in this process (Fig. S6B, S8A, and S8D-E).

B

A

D

E

Thank you once again for your constructive feedback.

4. It would be nice to show if there is any phenotype on beige adipocytes after cold exposure.

Response: Thank you for your thoughtful comments and valuable suggestion. We appreciate your interest in exploring the potential phenotype of beige adipocytes following cold exposure. As shown in Fig. 1M-N, we observed that cold challenge led to the death of all *Ythdc1*-BKO mice within 4 hours of exposure, indicating that *Ythdc1* deletion in brown adipose tissue (BAT) significantly impairs thermogenesis. Due to this acute lethality, performing a chronic cold challenge to induce beige adipocytes was not feasible in this model.

Instead, we utilized the β 3-adrenergic agonist CL316,243 to induce beige adipocytes, as depicted in Fig. 1O-P. Our data demonstrate that the formation of beige adipocytes (the browning of white adipose tissue) is dramatically impaired in *Ythdc1*-BKO mice, providing further insight into the role of BAT YTHDC1 in the regulation of beige adipocyte formation.

We hope this clarifies our experimental approach. Thank you again for your constructive feedback.

Minor: The figure legend is too long. Some content could be moved to the method section.

Response: Thank you for your comment. We understand your concern regarding the length of the figure legend. However, we have found that many scientists and readers prefer to review figures and figure legends directly rather than referring to the Methods section. Therefore, we would prefer to retain this information in the figure legend, provided the journal allows. We hope this approach is acceptable, and we are happy to make further adjustments if needed.

Dear Dr Chen,

Thank you for submitting your revised manuscript (EMBOJ-2024-119515R) to The EMBO Journal, as well for your patience with our response. Your amended study was sent back to the three referees for their scientific re-evaluation, and we have received detailed comments from two of them, which I enclose below. Please note that while referee #3 was at this time not able to reassess your amended work, we have asked referee #1 to also look into your response to this expert, in order to have this well covered. As you will see, the experts state that the work has been substantially enhanced by the revisions and they are now broadly in favour of publication.

Thus, we are pleased to inform you that your manuscript has been accepted in principle for publication in The EMBO Journal.

We now need you to take care of a number of issues related to formatting and data presentation as detailed below, which should be addressed at re-submission.

Please contact me at any time if you have additional questions related to below points.

As you might remember from previous experience, every paper at the EMBO Journal now includes a 'Synopsis', displayed on the html and freely accessible to all readers. Besides a 'model' figure the synopsis includes 2-5 one-short-sentence bullet points that summarize the article. I would appreciate if you could provide these bullet points.

Thank you for giving us the chance to consider your manuscript for The EMBO Journal. I look forward to your final revision.

Again, please contact me at any time if you need any help or have further questions.

Best regards,

Daniel Klimmeck

>> Author Contributions: Remove the author contributions information from the manuscript text. Note that CRediT has replaced the traditional author contributions section as of now because it offers a systematic machine-readable author contributions format that allows for more effective research assessment. and use the free text boxes beneath each contributing author's name to add specific details on the author's contribution.

More information is available in our guide to authors.
<https://www.embopress.org/page/journal/14602075/authorguide>

>> Adjust the title of the 'Competing Interests' section to 'Disclosure and Competing Interests Statement' and move after Acknowledgements.

>> Correct order of manuscript sections: Abstract / Keywords / Introduction / Results / Discussion / Methods / Data Availability / Acknowledgements / Disclosure and competing interests statement // References / Figure legends / Tables and their legends / Expanded View Figure legends

>> Limit the title length to maximally 100 characters (incl. spaces).

>> References: adjust reference format to EMBO Journal format, 10 authors et al. .

>> Appendix file with ToC: the file with suppl. information should be renamed "Appendix" and uploaded as a PDF. Please rename the suppl. tables "Appendix Table S1" etc., and the suppl. figures "Appendix Figure S1". Please add a table of contents, including page numbers.

>> Dataset EV Legends: should be renamed "Dataset EV1" and a legend should be added in a separate tab/worksheet.

>> Funding: please enter all funding information also into the list of funders in our online system.

>> Add a Reagents and Tools table to the Methods section, as a separate file using the existing template in the Guide For Authors, listing key reagents, experimental models, software and relevant equipment.

>> Data availability section: remove referee tokens and make sure the GEO dataset is made publicly accessible. Remove the GEO reference from the 'RNAseq' section. Rename the 'Data available' section to 'Data availability' section.

>> Please minimise textual overlap with your 2020 study (Wang et al; PMID: 32245957)) in the results section.

>> Consider additional changes and comments from our production team as indicated below:

- Figure legends:

1. Please note that the exact p values are not provided in the legends of figures 1B, F, G, J, L, M, P; 2E, F, H, I, J, L; 4B, C, D, E, F, G I-L; 5A, C, D, F; 6B, C, E; 7A, B, C, G, I, N; 8A, B, E, F, H.
2. Please indicate the statistical test used for data analysis in the legend of figure 3A
3. Please note that information related to n is missing in the legends of figures 3A, 5A

We realize that it is difficult to revise to a specific deadline. In the interest of protecting the conceptual advance provided by the work, we recommend a revision within 3 months (8th Jul 2025). Please discuss the revision progress ahead of this time with the editor if you require more time to complete the revisions.

Referee #1:

The authors satisfactorily addressed the points brought up in the previous review and I judge this revised manuscript acceptable for publication.

Additional re-comment Referee #1 on authors' response to referee #3:

Although not perfect, the authors addressed referee #3's concerns reasonably.

Referee #2:

The authors have addressed my comments made during the initial review. Consequently, I feel it is appropriate to accept the manuscript.

The authors addressed the remaining editorial issues.

Dear Dr Chen,

Thank you for submitting the revised version of your manuscript. I have now evaluated your amended manuscript and concluded that the remaining minor concerns have been sufficiently addressed.

I am thus pleased to inform you that your manuscript has been accepted for publication in the EMBO Journal.

On a different note, I would like to alert you that EMBO Press offers a format for a video-synopsis of work published with us, which essentially is a short, author-generated film explaining the core findings in hand drawings, and, as we believe, can be very useful to increase visibility of the work. Please see the following link for representative examples and their integration into the article web page:

<https://www.embopress.org/doi/full/10.15252/emj.2019103932>

Best regards,

Daniel Klimmeck

Daniel Klimmeck, PhD
Senior Editor
The EMBO Journal
EMBO
Postfach 1022-40
Meyerhofstrasse 1
D-69117 Heidelberg
contact@embojournal.org